# Emergent Approaches to Efficient and Sustainable Polyhydroxyalkanoate Production

**DOI:** 10.3390/molecules26113463

**Published:** 2021-06-07

**Authors:** Dattatray K. Bedade, Cody B. Edson, Richard A. Gross

**Affiliations:** 1Center for Biotechnology and Interdisciplinary Studies, Rensselaer Polytechnic Institute, 110 8th Street, Troy, NY 12180, USA; bedadedattu03@gmail.com; 2New York State Center for Polymer Synthesis, Department of Chemistry and Chemical Biology, Rensselaer Polytechnic Institute, 110 8th Street, Troy, NY 12180, USA; edsonc@rpi.edu

**Keywords:** Polyhydroxyalkanoate, PHA, production, high cell density cultivations, productivity, downstream processing

## Abstract

Petroleum-derived plastics dominate currently used plastic materials. These plastics are derived from finite fossil carbon sources and were not designed for recycling or biodegradation. With the ever-increasing quantities of plastic wastes entering landfills and polluting our environment, there is an urgent need for fundamental change. One component to that change is developing cost-effective plastics derived from readily renewable resources that offer chemical or biological recycling and can be designed to have properties that not only allow the replacement of current plastics but also offer new application opportunities. Polyhydroxyalkanoates (PHAs) remain a promising candidate for commodity bioplastic production, despite the many decades of efforts by academicians and industrial scientists that have not yet achieved that goal. This article focuses on defining obstacles and solutions to overcome cost-performance metrics that are not sufficiently competitive with current commodity thermoplastics. To that end, this review describes various process innovations that build on fed-batch and semi-continuous modes of operation as well as methods that lead to high cell density cultivations. Also, we discuss work to move from costly to lower cost substrates such as lignocellulose-derived hydrolysates, metabolic engineering of organisms that provide higher substrate conversion rates, the potential of halophiles to provide low-cost platforms in non-sterile environments for PHA formation, and work that uses mixed culture strategies to overcome obstacles of using waste substrates. We also describe historical problems and potential solutions to downstream processing for PHA isolation that, along with feedstock costs, have been an Achilles heel towards the realization of cost-efficient processes. Finally, future directions for efficient PHA production and relevant structural variations are discussed.

## 1. Introduction

Polyhydroxyalkanoates (PHAs) are a class of biopolymers produced as intracellular energy/carbon storage materials that also possess versatile material properties. PHA was first discovered in *Bacillus megaterium* as granular inclusion bodies by the French scientist Lemoigne. Later these granular bodies were extracted and identified as poly (3-hydroxybutyrate) (PHB) [1]. PHAs remain of high interest as potential substitutes to conventional plastics in numerous fields of application due to their widespread applicability in various fields such as food packaging, agriculture, tissue-engineering scaffolds, bioresorbable implants and for drug delivery [2]. To date, various PHAs and their copolymers have been isolated from different bacterial species. More than 150 constituent repeat units have been reported to have been incorporated as PHA units along chains [3,4]. The monomer composition of PHAs can be altered so that the polymers have tailored physicochemical and mechanical properties [5]. 

PHAs are classified by the chain length of 3-hydroxyalkanoate (3HA) repeat units. PHAs with short chain-length repeat units (scl-PHA) contain primarily 4 and 5 carbon atoms in repeat units (e.g., 3-hydroxybutyrate [3HB] and valerate units [3HV]). In contrast, PHAs that consist of medium-chain-length PHA (mcl-PHA) have repeat units with chain lengths of 6-14 carbon atoms (e.g., poly [3-hydroxynonanoate]) [6]. The structure of the scl- and mcl-PHA are presented in Figure 1. 

Principal microbial producers of mcl-PHAs include *Pseudomonas* sp. Characteristic features of mcl-PHAs are that they are soft, ductile materials due to their low glass transition temperatures and crystallinities [7,8]. P(4-hydroxybutyrate) (P4HB), that has an extended methylene group between carbonyl and oxygen moieties, is just one example of a comonomer in 3HB copolymers that has shown promising material properties. Increasing the content of 4HB units in poly(3HB-*co*-4HB) tends to increase copolymer ductility while decreasing its melting point and % crystallinity. Comprehensive reviews have been published that elaborate on the effects of PHA composition on its physico-mechanical properties and it is not our intent herein to recapitulate this information [2,9,10,11]. 

A major bottleneck in the commercialization of PHAs is the high cost of production where metrics such as carbon conversion yield (g/g), titer or volumetric yield (g/L) and productivity (g/L/h) are critically important [12]. In addition to production metrics, low cost downstream processing methodologies and PHA manufacturing that meets cost-performance requirements have remained challenging. This has inspired researchers to work towards increasing PHA fermentation and downstream processing efficiency to reduce the overall cost [13,14,15,16].

To date, more than 300 bacterial species were identified that produce PHA under aerobic/anaerobic conditions and extremophiles such as halophiles that provide genetic diversity and diverse production conditions [17,18].

PHA is accumulated in the form of granules (size range from 0.2–0.5 μm) in the cytoplasm of bacteria. Based on PHA production capability, bacteria are classified as to: (1) whether PHA predominantly accumulates during the stationary phase under nutrient (e.g., nitrogen, phosphorous, oxygen and magnesium) limiting conditions or (2) PHA is formed during the growth phase without nutrient limitation [2]. *Pseudomonas putida, Pseudomonas oleovorans* and *Ralstonia eutropha* require nutrient limiting conditions for PHA production whereas recombinant *Escherichia coli* and *Alcaligenes latus* do not [19]. 

PHA synthesis occurs by the consecutive action of a *β*-ketoacyl-CoA thiolase (*PhaA*), acetoacetyl-CoA reductase (*PhaB*), and P(3HB) polymerase (*PhaC*) (Figure 2). 

The *pha**CAB* operon encodes these enzymes and the upstream promoter of *phaC* transcribes the complete operon. The biosynthetic pathway involved in biosynthesis of mcl-PHA is displayed in Route B (Figure 2). Indeed, significant efforts have been made to engineer these enzymes with enhanced activity and vary their substrate specificity. Detailed discussions of metabolic and protein engineering of PHA biosynthetic enzymes are reviewed elsewhere [20].

For production, achieving high PHA cell contents during high cell-density cultivations (HCDC) is a key objective that leads to high product titers. HCDC was first established using yeasts that fabricate bioethanol and single-cell proteins [21]. Later, HCDC was explored for production in high titers of antibiotics and PHA using mesophilic strains [6,22]. HCDC processes are favored over low-density processes due to their advantages such as reduction of culture volume and residual liquids, reduced cost of production and lower capital investment [23,24]. Continuous and fed-batch cultivations are crucial operation modes used to attain HCDC of bacteria for PHA production. While cell dry weight (CDW) above 50 g/L are considered as high for production of recombinant proteins [21,25], cell densities and residual biomass above 100 g/L and 30–40 g/L, respectively, are considered as HCDC for PHA production [15,26,27,28,29,30]. 

Figure 3 illustrates the potential sustainable production of PHAs. This figure encompasses much of what will be discussed in this review. To meet sustainability metrics measured by life-cycle analysis, it is critical that carbon sources are derived from non-food sources such as lignocellulose; conversions of feedstocks to products occurs with high carbon conversion efficiencies; downstream processing is achieved with minimal process steps, inputs of chemicals (i.e., enzymes), and energy utilization; and, after use, the products are disposed of in bioactive environments such as composts or are degraded chemically or enzymatically to building blocks that can be re-used. 

In this review article, we summarize HCDC methods for the biosynthesis of scl- and mcl-PHA and associated strategies that lead to increased productivity. We will also discuss approaches such as nutrient limitation, genetic and metabolic engineering, use of mixed culture and renewable carbon sources for enhancement of PHA production efficiency. Recent developments on cost- effective downstream processing are also discussed herein.

## 2. Modes of Operations for Production of Polyhydroxyalkanoates (PHAs) in High Cell Density Cultivations

The productivity of PHAs is dependent on many factors such as the bacterial strain, carbon/nitrogen ratio, pH, temperature, cultivation time, and presence of micro and macro nutrients [31]. For optimization of PHA yield, different fermentation strategies such as batch, fed-batch and continuous processes have been used. Figure 4 provides a diagram of the overall PHA production pathway with options of alternative processes. Fermentation processes can be separated into two categories: continuous or discontinuous processes. The upper half of the figure illustrates discontinuous processes: batch fermentations, repeated fed-batch and fed-batch coupled with cell recycling. The bottom half of the figure illustrates continuous processes: single-stage chemostat, two-stage chemostat and a multi-stage bioreactor cascade. Unlike batch fermentations, continuous processes maintain static fermentation parameters. The use of multi-reactor fermentation strategies, especially multi-stage cultivation systems, are recommended to obtain high yields of PHA. R1–R5: Five continuous stirred tank reactors. F1, F3, F5, F7 and F9: Feed streams for supply of nutritional medium to the bioreactors R1, R2, R3, R4 and R5, respectively. F2, F4, F6 and F8: Continuous transfer of fermentation broth to the subsequent reactors. F10: Outlet stream containing final product. Microbial cell growth occurs in R1, whereas PHA accumulation takes place in R2–R5.

### 2.1. Batch Cultivations

In batch cultivations, the carbon/nitrogen sources are added to the system at time zero of the cultivation. That is, during batch cultivations, additional nutrients are not added during the course of the fermentation [32]. Generally, batch fermentation processes result in low PHA yields which is partly attributed to PHA degradation that occurs at the later during cultivations [18]. Singh et al., investigated P3HB production from industrial sugar waste using *Bacillus subtilis* NG220. Cultivations resulted in 10.22 g/L of biomass that contained 51.8% PHA [33]. Rai et al., investigated the batch production of P(3-hydroxyoctanoate) from *Pseudomonas mendocina* using sodium octanoate. This resulted in very low biomass yield of 0.8 g/L with 31% of homopolymer production using sodium octanoate as sole carbon source [34]. 

### 2.2. Fed-Batch Fermentations

The fed-batch fermentation method was originally developed in the early 1900s by yeast producers such as cultivation of *Saccharomyces cerevisiae* [35]. Later, this concept was applied for production of antibiotics, amino acids, microbial cells, enzymes, growth hormones, vitamins, organic acids and PHA [36]. Fed-batch processes are extensively used for industrial fermentations due to its distinct advantages over other modes of operation of a bioreactor. In a fed-batch process, cells are grown under continuous feeding of carbon sources and essential nutrients at a certain rate until the desired cell density is attained. The feed solution containing carbon sources and essential nutrients maintains a specific growth rate that reduces by-product formation. 

There are two types of fed-batch cultivations: product formation that is either growth-associated or that occurs under non-growth-associated conditions. PHA production is usually carried out in two phases, first, the growth phase is conducted such that cells reach the desired biomass. Polymer production occurs in the second phase in which all essential nutrients required for production are fed to the bioreactor [37]. In most cases, the second phase is conducted where an essential nutrient, such as phosphorus, nitrogen, oxygen and sulfur, is at a limiting concentration such that metabolic pathways supporting growth are suppressed and the cells respond by focusing their efforts on PHA accumulation to store carbon and energy. Fed-batch cultivations must have suitable strategies to supply carbon sources and other nutrients. Under circumstances where a carbon source for PHA becomes limiting, the rate of PHA degradation via a PHA depolymerase increases.

The use of fed-batch cultivations has proved valuable to achieve PHA production under HCDC conditions [29,38,39]. Cultivations of *Cupriavidus necator* under fed-batch HCDC has been a target of interest by numerous research groups and the results of this work are displayed in Table 1.

Mcl-PHAs have also been produced by HCDCs and Table 1 summarizes the results of published work. The pH stat and DO (dissolved oxygen) stat involves maintaining the pH and DO at certain levels during the fermentation process. Lee et al., reported a strategy for achieving HCDCs using *P. putida* as the microbial catalyst and oleic acid as the carbon source [15]. Oleic acid feeding was controlled by a pH-stat during the growth phase and a DO-stat in the polymer production phase. This results in a total biomass, %-PHA in the CDW and overall productivity were 141 g/L, 51% and 1.9 g/L/h, respectively [15]. 

A two-stage fed-batch HCDC of *P. putida* KT2440 using glucose and nonanoic acid was reported by Davis et al. Cells were grown on glucose in biomass accumulation phase, and nonanoic acid was fed in the PHA production phase. Using this two-stage feeding strategy, *P. putida* KT2440 accumulated a total biomass, %-PHA CDW content and overall productivity of 102 g/L, 32% and 0.98 g/L/h, respectively [26]. The authors claimed that, this two-stage feeding strategy resulted in the highest ever reported value of biomass for a *P. putida* strain. 

While cell concentration increases during a fed-batch fermentation, one can impose a slowly decaying specific growth to attain high cell densities while preventing other perturbations that could result from rapid decreases in the specific growth rate. Maclean et al., reported a decaying exponential feeding of nonanoic acid during a *P. putida* KT2440 cultivation to form an mcl-PHA. A linear and quadratic decaying exponential feeding rate were used to control biomass accumulation and, subsequently, to control the oxygen uptake rate of the cells. The latter strategy resulted in in a total biomass, CDW, % PHA content and overall productivity of 109 g/L, 63% (i.e., 69 g/L PHA) and 2.3 g/L/h, respectively [27]. The larger final biomass concentration and mcl-PHA content is explained by the directly correlation between the highest rates of cell formation and oxygen uptake. 

PHA-producing organisms (e.g., *R. eutropha*) can use acetic, propionic and butyric acids as primary substrates for both biomass accumulation and PHA formation. Propionate/propionic acid introduced in culture media serves as a precursor for the formation of 3HV units in P(3HB-*co*-3HV). The incorporation of 3HV units increases the ductility while decreasing the copolymer melting point relative to P3HB. Huschner et al., reported fed-batch cultivations of *R. eutropha* that functioned to decrease the toxicity of organic acid substrates. The rate of organic acid feeding was pO_2_-dependent. This approach resulted in highly reproducible cultivations that reached a total biomass, %-P(3HB-*co*-5.6 mol%3HV) content and overall productivity of 112 g/L, 83% and >2 g/L/h, respectively [28]. 

Yamane et al., reported a fed-batch cultivation of *A. latus* where sucrose, inorganic elements and an ammonia solution were fed into a bioreactor by the pH-stat method. This work highlighted the importance of carbon and nitrogen feed concentrations in obtaining high PHA productivity. Consumption rates were used to inform when to supply carbon/nutrient sources that enabled nutrient concentrations to be maintained at nearly constant levels throughout cultivations. The feeding solutions were supplied based on their consumption rates, thereby maintaining the nutrient concentrations at nearly constant levels during the fermentation. This approach resulted in a total biomass, PHA content in the CDW and overall productivity of 143 g/L, 50% and 3.97 g/L/h, respectively [40]. 

PHB production by *C. necator* DSM 545 from glucose by a two-phase fed batch cultivation was reported by Mozumder et al. The first phase was dedicated to biomass production after which, in the second phase, a specific PHB production rate and nutrient-limiting conditions induced PHB formation. Process optimization led to a total biomass, PHB content in the CDW and overall productivity of 164 g/L, 76.2% and 2.03 g/L/h, respectively [41]. 

*Burkholderia sacchari* was identified as an efficient micro-organism for P3HB production. Sucrose from sugarcane was the primary carbon source while *γ*-butyrolactone (GBL) was used as a co-substrate for the formation of 4HB units. P3HB formed in the absence of GBL reached a PHB concentration and overall productivity of 36.5 g/L and 1.29 g/L/h, respectively [42]. However, addition of GBL results in P(3HB-*co*-1.6 mol%4HB) at a concentration and volumetric productivity of 54 g/L and 1.87 g/L/h, respectively [42]. 

Haas et al., reported the use of a membrane bioreactor and HCDC for P3HB formation. PHB was formed by continuously feeding of a synthetic medium with 50 g/L glucose to *C. necator*. This strategy resulted in a total biomass, P3HB content in the CDW and overall productivity of 148 g/L, 76% and 3.10 g/L/h, respectively [43]. The bacterium rapidly consumed fed sugar resulting in low contents in the final medium. 

Cell recycle has been successfully implemented in fed-batch and continuous cultures resulting in HCDC and efficient PHA formation [44,45]. Ienczak et al., demonstrated that by coupling repeated fed-batch cultivations with cell recycle, HCDC of *C. necator* DSM 545 was achieved from glucose and fructose (90 g/L) [45]. Culture media depleted of nutrients was removed from the bioreactor without loss of cells during recirculation. The results showed that total biomass, P3HB yield, P3HB CDW content and overall productivity reached 61.6 g/L, 42.4 g/L, 68.8% and 1 g/L/h, respectively [45]. It was noteworthy that, by this cultivation strategy, carbon source concentration about 7-fold below that used in other studies was effectively converted to product. This is particularly meaningful for waste feedstocks that contain low sugar concentrations. Schmidt et al., used *C. necator* DSM 545 for P(3HB-*co*-3HV) by external cell recycling under a production phase where nitrogen limitation was imposed. The glucose concentration fed to the culture medium simulated that often found in agro-industrial wastes (90 g/L). To induce 3HV formation, propionic acid was used as a co-substrate. The results showed that the total biomass, PHB content in the CDW and overall productivity reached 80 g/L, 73% and 1.24 g/L/h, respectively [46]. 

Rodríguez-Contreras et al., substituted glucose for glycerol during the production of P3HB by *C. necator* and *B. sacchari*. The maximum biomass, P3HB CDW content and productivity for these strains were 68.6 g/L, 64.6%, 0.76 g/L/h and 43.8 g/L, 10.2%, 0.08 g/L/h, respectively [47]. The isolated bacterium *Zobellella denitrificans* MW1 that possess a high capability to accumulate PHA from glucose, was assessed in a pilot scale reactor (42 L) for P3HB formation from glycerol. Using fed-batch cultivation, the optimized feeding strategy of glycerol and sodium chloride results in 81.2 g/L total biomass, 67% PHA content and 1.09 g/L/h productivity [48]. 

Chanprateep et al., reported the formation of P(3HB-*co*-4HB) during fed-batch HCDC of *C. necator* A-04. Fructose served as the carbon sources for biomass accumulation while 1,4 butanediol functioned as a 4HB precursor. The authors varied the molar ratios of fructose to 1,4-butanediol and, consequently, altered the composition and productivity of P(3HB-*co*-4HB) formation. The culture in which P(3HB-*co*-38% mol 4HB) was produced reached a total biomass, PHA CDW content and overall productivity of 112 g/L, 65% and 0.76 g/L/h [49].

Le Meur et al., reported increased scl-PHA productivity by recombinant *E. coli*. Glycerol served as the carbon source for biomass accumulation while 4HB functioned as a precursor for 4HB units. Pulse, linear and exponential feeding strategies were evaluated, the exponential feeding of glycerol and butyric acid was found to be highly reproducible and results in biomass, PHA CDW and overall productivity were 43.2 g/L, 33% and 0.207 g/L/h [50]. 

Stanley et al., reported pH-based and pulse feeding strategies to improve PHA yields during *Halomonas venusta* cultivations. Usually, the fermentation broth pH moves toward lower values during microbial growth due to the production of organic acids. In contrast, the medium pH increases under carbon source limiting conditions due to the production of ammonium ions during protein catabolism. Consequently, the feed pump was altered such that, in the event that the pH increased above a set value (7.05 in the current study), this cues the automated feeding of the carbon source. Using this pH-based strategy, the authors reported a total biomass and PHA CDW content of 66.4 g/L and 39%, respectively [51]. Also, they found that, when the maximum utilization of glucose was reached, a single pulse (100 g/L in this study) was used to increase the available glucose in the bioreactor. The single pulse feeding approach resulted in an accumulated biomass and %-PHA content of 37.9 g/L and 88%, respectively [51]. The increase in PHA content in pulse feeding could be due to increased glucose flux towards PHA synthesis. *B. sacchari* IPT 189 was cultivated for P(3HB-*co*-3HV) production using sucrose and propionic acid and a two-stage bioreactor process [52]. During the first stage, a balanced culture medium was used for growth up to sucrose exhaustion. The second stage constituted feeding sucrose/propionic acid solution to the bioreactor. The sucrose/propionic acid ratio was varied while the feed flow rate was kept constant. The results showed that, by increasing the ratio of sucrose to propionic acid to 30:1, the %-PHA [P(3HB-*co*-10 mol%3HV)] cell content and productivity reached 60% and 1.04 g/L/h, respectively [52].

In pursuit of HCDC for P(3HB-*co*-3HV) production by *Aeromonas hydrophila* 4AK4, Chen et al., employed the cofeeding of glucose/lauric acid in a two-stage fermentation. Lauric acid pulsed feeding results in 20 g/L residual carbon source concentration and a total biomass, %-PHA CDW content and productivity of 50 g/L, 50% and 0.54 g/L/h, respectively [53]. 

Blunt et al., reported an oxygen-limited fed-batch cultivation process for enhanced productivity of mcl-PHAs using *P. putida* LS46. They used octanoic acid as the carbon source and a bench-scale (7 L) bioreactor. The resulting total biomass, %-PHA CDW content and productivity reached 29 g/L, 61% and 0.66 g/L/h, respectively [54]. The relatively low biomass accumulation may be due to the toxicity of octanoic acid to *P. putida* LS46 cells. Gao et al., conducted a fed-batch cultivation of *P. putida* KT2440 mcl-PHA production with a co-feed mixture of decanoic and acetic acids [55]. Acetic acid functioned to prevent decanoic acid crystallization. To identify co-feed ratios that would result in higher mcl-PHA yields, different ratios of decanoic acid/acetic acid/glucose was used. With the optimized ratio (5:1:4), the total biomass, %-PHA CDW content and overall productivity reached 75 g/L, 74% and 1.16 g/L/h, respectively [55].

Sun et al., reported the formation of mcl-PHA by *P. putida* KT2440 by co-feeding glucose and nonanoic acid during a carbon-limited fed-batch cultivation. Exponential and, thereafter, linear feeding of 1:1 (*w*/*w*) nonanoic acid: glucose resulted in a total biomass, %-PHA CDW content and overall productivity of 71 g/L, 56% and 1.44 g/L/h, respectively [56]. Cerrone et al., reported the HCDC of *P. putida* CA-3 by a two-stage fermentation of co-substrates decanoic and butyric acid [57]. To enhance the mcl-PHA volumetric productivity, the cells were initially grown on butyric acid (biomass growth phase) and, subsequently, during the PHA production stage, the carbon source used was a mixture of butyric and decanoic acid (20:80 *v*/*v* ratio). This strategy resulted in a total biomass, %-PHA CDW content and overall productivity reached 71.3 g/L, 65% and 1.63 g/L/h respectively [57]. 

Sun et al., reported the HCDC of *P. putida* KT2440 for mcl PHA formation from nonanoic acid. An exponential growth rate (μ = 0.15 h^−1^) under nonanoic acid-limited conditions resulted a total biomass, %-PHA CDW content and overall productivity of 70 g/L, 75% and 1.11 g/L/h, respectively [58]. However, by increasing the exponential feed rate to μ = 0.25 h^−1^, the overall productivity increased (1.44 g/L/h), however, the biomass (56 g/L) and mcl-PHA content (67%) decreased due to the higher oxygen demand [58]. Diniz et al., studied different feeding strategies such as pulse feed followed by constant feed, and exponential feed to produce mcl-PHA using *P. putida* IPT 046 [59]. The exponential feeding strategy results in total biomass of 40 g/L with 21% mcl-PHA content. However, under phosphate limitation, biomass accumulation, the CDW content of mcl-PHA and overall productivity reached 50 g/L, 63% and 0.8 g/L/h, respectively [59]. 

Cultivation of *P. oleovorans* ATCC 29347 was conducted under pH-stat fed-batch conditions using octanoic acid as the feedstock. The resulting total biomass, %-PHA in the CDW and overall productivity were 63 g/L, 62% and 1 g/L/h [60]. Kim et al., reported *P. putida* BM01 cultivation by a two-stage fed-batch process. These workers co-fed glucose and octanoate during both biomass growth and PHA production. This strategy resulted in a total biomass, %-PHA in the CDW and overall productivity of 55 g/L, 66% and 0.90 g/L/h, respectively [61]. 

With the objective of improving the distribution of both carbon and energy, Andin et al., coupled *P. putida* KT2440 growth and mcl-PHA production from fatty acids [62]. Experimental data validated a model that describes the energy flux distribution and carbon utilization in *P. putida* KT2440 during the simultaneous processes of growth and PHA formation. This approach explored the possibility of shifting available carbon and energy to PHA formation during the production phase. The resulting fed-batch culture had a total biomass, %-PHA in the CDW and overall productivity of 125.6 g/L, 54.4% and 1.01 g/L/h, respectively [62]. Thus, one can couple PHA formation and growth when substrate catabolism occurs via *β*-oxidation. 

Dey and Rangarajan reported a HCDC of *C. necator* (MTCC 1472) on sucrose for P3HB formation by a fed batch fermentation [63]. Under nitrogen limited fed-batch cultivation, the concentration during feeding of sucrose was varied from 100–200 g/L. The total biomass, %-P3HB in the CDW and overall productivity reached 38 g/L, 62% and 0.58 g/L/h, respectively, at a dilution rate of 0.046 h^−1^ and by feeding a 200 g/L sucrose solution [63]. The authors claimed their approach provided an economically attractive route to PHB production. 

To maximize the P3HB accumulation rate of *Azohydromonas lata* DSM 1123, Penloglou et al., adopted an intensified fed-batch process based on a model. The models were validated to determine optimal feeding and operating conditions that optimize P3HB productivity. By a continuous feeding strategy under non-limiting nitrogen conditions, a maximum PHB CDW content of 94% overall productivity of 4.2 g/L/h, was reported [64]. However, the authors did not provide the value of the total biomass concentration accumulation. 

### 2.3. Continuous Culture

By this technique, the rate of microbial growth is constant under steady-state conditions. A continuous cultivation process that runs at high specific growth rates can provide high productivities. Furthermore, continuous cultivations are desirable since they substantially decrease the frequency of bioreactor shutdown and cleaning operations. Also, continuous cultivation processes circumvent wash-out even at high dilution rates. This can lead to high productivity and concentrations of the product. To minimize the disruption of normal microbial cellular behavior, continuous or semi-continuous processes can be implemented in place of batch process. Continuous cultivation processes are characterized by the continuous addition at a constant flow rate of fresh media to the bioreactor which provides the cells with fresh nutrients. To keep the bioreactor working volume constant, products and effluents are continuously removed. Representative outcomes of PHA production under continuous HCDC conditions are displayed in Table 2.

Jung et al., reported a continuous two-stage process by which *P*. *oleovorans* converts *n*-octane to mcl-PHA. Two stage fermentations offer the opportunity to focus on biomass accumulation in the first bioreactor and PHA accumulation in a second bioreactor. In the first (D1) and second (D2) bioreactors, the dilution rate were 0.21 h^−1^ and 0.16 h^−1^, respectively. These conditions resulted in a total biomass, %-mcl-PHA content in the CDW and overall productivity of 18 g/L, 63% and 1.06 g/L/h, respectively [65]. 

Atlic et al., reported a continuous cultivation of *C. necator* for P3HB production from glucose [66]. The multistage reaction system consisted of five bioreactors in series. The first bioreactor functioned for biomass accumulation; thereafter, the fermentation broth was continuously fed into subsequent reactors for P3HB production under nitrogen limiting conditions. The dilution rate (0.139 h^−1^) for the cascade experiment was substantially higher when compared to the corresponding 2-stage process (0.075 h^−1^) since the authors assumed that the five-reactor series would have a relatively higher product throughput. Upon reaching steady state conditions, the total biomass, %-P3HB CDW content and overall productivity reached 81 g/L, 77% and 1.85 g/L/h, respectively [66]. This work highlights how, by adopting a continuous process using a series of bioreactors, high product titers and productivity can be achieved. 

As above, Horvat et al., reported a continuous cultivation of *C. necator* for P3HB production from glucose [67]. The multistage reaction system consisted of five bioreactors in series. For the first bioreactor in the cascade, modelling was based on maintaining a nutrient balanced system with continuous biomass production. The second bioreactor adopted a model for process control using two substrates. Control of the next three bioreactors aimed to achieve high P3HB formation under nitrogen limitation with continuous glucose feeding. They reached a total biomass, %-P3HB in the CDW and overall productivity of 80 g/L, 77% and 2.14 g/L/h, respectively [67]. Du et al., adopted a continuous two-stage cultivation where the first and second bioreactors were optimized for biomass and P3HB accumulation, respectively [68]. P3HB formation in the second bioreactor was under nitrogen limiting conditions. After optimization of the dilution rates (0.075 h^−1^) and carbon source (50 g/L in first stage and 500 g/L in second stage), they reported a total biomass, %-P3HB content in the CDW and an overall productivity of 50 g/L, 73% and 1.23 g/L/h, respectively [68]. 

To study the kinetics of P3HB synthesis, Du et al., performed a continuous cultivation of *R. eutropha* containing two bioreactors in series [69]. In the first bioreactor, *R. eutropha* cells were cultivated under limiting glucose (feeding solution concentration 50 g/L) conditions. In the second bioreactor, P3HB accumulation occurs with excess carbon source (feeding solution concentration 500 g/L) and limiting nitrogen conditions. The specific P3HB production rate was dependent on the C/N molar ratio such that, the C/N ratio of 30 in PHB production phase gave optimal results: biomass accumulation reached 32.6 g/L and %-PHB content in dried cells was 75% [69]. Khanna and Srivastava explored the formation of P3HB formation by *Wautersia eutropha* NRRL B-14690 under continuous cultivation conditions [70]. P3HB formation was induced by imposing nutrient limiting conditions. Minimal P3HB formed during the exponential growth phase. P3HB formation in the second stage was increased by low dilution rates. Under these conditions the authors reported a total biomass, %-P3HB in the CDW and overall productivity of 49 g/L, 51% and 0.42 g/L/h, respectively [70]. 

Egli et al., used chemostat culture conditions to investigate PHA formation by *P. putida* GPo1. This work revealed that, when both carbon and nitrogen simultaneously limit growth, PHA formation can occur. Under these conditions, studies were conducted to determine how the C/N ratio, substrate type and the cell growth rate affected product formation. A correlation was found between increased *P. putida* GPo1 PHA formation and prolonged carbon and nitrogen limiting cultivation conditions [71]. Similarly, Zinn et al., reported on a cultivation of *R. eutropha* DSM 428 under both nitrogen and carbon limiting conditions. The carbon sources used were butyric and/or valeric acid while ammonium served as the source of nitrogen. This strategy results in a cellular PHA content of 40% [72]. Unfortunately, the authors did not provide sufficient information to calculate the cell concentration and productivity. 

Yu et al., reported the continuous production by *R. eutropha* of P(3HB-*co*-3HV) using glucose and sodium propionate as co-substrates. Increased molar fractions of HV units in the final product resulted by increasing the relative concentration of sodium propionate in the feed. This resulted in a total biomass, P(3HB-*co*-60 mol%3HV) CDW content and an overall productivity of 8 g/L, 30% and 0.045 g/L/h, respectively [73]. While increase in the sodium propionate concentration correlated with higher copolymer 3HV content, 3HV can inhibit *R. eutropha* growth decreasing both the biomass and P(3HB-*co*-3HV) formation. A continuous cultivation of *P. putida* KT2442 for biosynthesis of mcl-PHA was studied by Huijberts and Eggink. Using oxygen limited continuous HCDC, the total biomass, mcl-PHA cell content and overall productivity reached 30 g/L, 23% and 0.69 g/L/h, respectively [74]. 

*Halomonas* sp. TD01 is highly tolerant to both high salt and pH conditions. This is advantageous since, such an environment is intolerant to other potential strains that pose contamination risks. Consequently, the rigors of processes normally used to maintain sterile conditions can be relaxed such that continuous and open fermentation processes can be used without concerns of contamination. In one example, an unsterile two-stage continuous cultivation for P3HB production by halophilic bacteria *Halomonas* TD01 was reported by Tan et al. [75]. *Halomonas* TD01 cells were cultivated on glucose in the first bioreactor for 2 weeks and, thereafter, the cells were transferred into the second bioreactor under nitrogen limiting conditions. While the continuous transfer of cultures from the first to the second bioreactor diluted the cells, the %-P3HB in the CDW remained at between 65–70% [75]. At 24 h, the first fermenter had a biomass of 40 g/L that contained 60% P3HB. These values were maintained during the entire cell growth period. In the second stage, the total biomass, %-mcl-PHA in the CDW and overall productivity reached 20 g/L, 65% and 0.26 g/L/h, respectively [75]. The low cell biomass in reactor 2 is a consequence of culture dilution while maintaining high P3HB content results from nitrogen limitation. These results highlight that *Halomonas* TD01 is an attractive cell bio-factory for P3HB accumulation since the culture conditions are highly amenable for commercial processes. However, further development of the organisms and process is needed to reach commercially viable PHA yields and productivities. 

Another approach to conduct semi-continuous fermentation is by cyclic fed-batch fermentations (CFBF) at high cell densities. CFBF is performed by partially removing culture broth with subsequent refilling of fresh medium to the bioreactor [76]. This approach circumvents an accumulation of toxic concentrations of by-products and corresponding increased culture volumes that occur during fed-batch fermentations [77]. As a result, CFBF has proved useful in reducing the impact of media chemical changes enabling thermophiles to reach high final biomass and product concentrations. Ibrahim and Steinbuchel reported the application of CFBF in a stirred tank reactor for HCD cultivation of the PHB accumulating thermophile *Chelatococcus* sp. Strain MW10. The aim was to develop energy-saving PHB production processes. Using this strategy, total biomass reached 115 g/L but the %-PHB in the CDW was relatively low (12%) [78]. Nevertheless, CFBF is attractive for thermophilic strains for the reasons described above as well as the relatively simple fermenter set up and ability to monitor by the withdrawal/refilling process. 

Karasavvas and Chatzidoukas, reported the modelling and dynamic optimization two continuous cascade bioreactors to optimize P3HB formation from sucrose by *Azohydromonas lata*. For the system at steady state they reported a total biomass and %-PHB CDW content of 20.52 g/L and 83.4%, respectively [79]. 

## 3. Effect of Nutrient Limitations on Yield of PHA

Nutrient limitation is a key strategy for PHA production processes. Different nutrients have different effects on cell metabolism, growth, and PHA production. PHA production under nutrient limitations is generally conducted by a two-stage fed batch cultivation in which PHA accumulation occurs primarily during the nutrient-depleted stage [15,59,61]. Nutrient limiting conditions is imposed by continuous feeding of essential nutrients while reducing the concentration of the growth limiting nutrient (i.e., nitrogen) to reach a desired C/N ratio [60,80,81]. Increased PHA cell contents under conditions that are nutrient limiting is a direct result of imposing constraints such that, available carbon sources are not used for biomass accumulation but, instead, for PHA synthesis [82]. For example, the rate of scl-PHA synthesis by *R. eutropha* is significantly increased under nitrogen and phosphorus limitations [83]. The effect of nutrient limitations on PHA synthesis is summarized in Table 3. 

Sun et al., (2007) reported simultaneous growth and accumulation of mcl-PHA using *P. putida* KT2440 where the rate of non-PHA biomass accumulation is below that of PHA biosynthesis [58]. Hence, PHA synthesis is not strictly associated with cell growth. Lee et al., reported mcl-PHA fed-batch production using *P. putida* KT2440 under phosphorous limitations that led to impressive mcl-PHA production. To increase PHA cell content, the initial phosphorus concentration in the feed was varied during fed-bath cultivations. By reducing the initial concentration of KH_2_PO_4_ from 7.5 to 4 g/L, the total concentration of total biomass and mcl-PHA cell content reached 141 g/L and 51.4% (i.e., mcl-PHA yield of 72.6 g/L), respectively, with an overall productivity of 1.91 g/L/h [15]. When the initial phosphate concentration was further reduced, the PHA content of the CDW remained unchanged but the overall productivity and concentration of PHA was reduced. These workers also provided a useful roadmap as to how the feeding strategy can be used to reach HCDC and productivity values. Furthermore, the time at which the nutrient limitation was applied significantly affected biomass accumulation, PHA cell concentration and productivity. The initial phosphorus concentration mainly affected the conversion efficiency of acetate to PHA. Ryu et al., reported *A. eutrophus* fed-batch cultivations under phosphorus limitation for P3HB production [29]. The dissolved oxygen (DO) concentration was used to control both the glucose feeding rate as well as to monitor its concentration. Variation in the glucose concentration was between 0-20 g/L. In addition, the influence of the initial phosphate concentration on P3HB formation was evaluated. At 5.5 g/L initial phosphorus concentration, the total biomass, mcl-PHA cell content and overall productivity reached 281 g/L, 82% and 3.14 g/L/h, respectively [29]. These results also stand out as highly impressive and provide valuable information on how high PHA yields can be attained. 

Shang et al., investigated the effect of the glucose feeding rate on the formation of P3HB by *R. eutropha* under phosphate limiting and fed-batch cultivation conditions. By sustaining the glucose concentration in the medium at 2.5 g/L, P3HB formation and cell growth were restricted by the carbon source shortage. However, by sustaining the concentration of glucose in the culture at 9 g/L, the total biomass, %-P3HB in the CDW and overall productivity reached an impressive 208 g/L, 67% and 3.1 g/L/h, respectively [39]. However, further increase of the glucose concentration in culture media to 16 g/L resulted in significant decreases in P3HB productivity.

Tu et al., evaluated the effect of phosphorus limitation on accumulation of PHA from thermally hydrolyzed sludge. Decrease in the phosphorus concentration from 127.6 to 1.35 mg/L resulted in an increase in PHA cell contents from 23 to 51% [84]. 

Wen et al., evaluated how nitrogen and phosphorus limitation effected PHA formation from acetate. The microbes in this study were from activated sludge. Ratios of C:P and C:N were varied to investigate the effect of nitrogen and phosphorus limitation, respectively. The maximum %-PHA in the CDW reached 59% at the C:N 125 and 37% under phosphorus limitation experiments [85]. However, the authors did not provide information on how nutrient limitations affected the total biomass accumulated and overall productivity.

Portugal-Nunes et al., investigated the effect of nitrogen availability on PHB accumulation in two recombinant strains of *S. cerevisiae* using xylose as the carbon source. However, nitrogen deficiency did not enhance PHB accumulation in *S. cerevisiae*. Instead, the highest PHB contents (2.7-fold increase) were obtained under excess of nitrogen [86]. 

Grousseau et al., reported cultivation of *C. necator* on a butyric acid/propionic acid co-feed during fed batch fermentation conditions. They discussed how the distribution of 3HB and 3HV monomers was influenced by the ratio of propionic and butyric acids in the feed. Decreasing 3HB with sustained 3HV formation occurred under phosphorus limited conditions. In fact, under these conditions the PHA formed consisted of nearly 100% 3HV units [87]. By feeding phosphorus, which sustained cell growth, and using propionic acid as the carbon source, the maximum 3HV content in the P(3HB-*co*-3HV) copolymer was 33% [87]. This was explained by the fact that, by imposing a phosphorus limitation, the decarboxylation of propionic acid decreased thereby maximizing 3HV production. By cofeeding butyric and propionic acid (1:2 molar ratio), the total biomass, %-P3HB in the CDW and overall productivity reached 65.9 g/L, 88% and 0.65 g/L/h, respectively [87]. Furthermore, by moving from a feedstock that consists of only propionic acid to one with a butyric acid co-substrate, metabolism of propionic acid to 3HV occurs at higher efficiency.

Da Cruz Pradella et al., reported the HCDC of *B. sacchari* 189 on sucrose in an airlift bioreactor. In a two-phase fed-batch fermentation experiment, nitrogen limitation induced P3HB biosynthesis. In phase one, a limited sucrose feeding regime resulted in 60 g/L biomass and a low (13%) %-P3HB content in the CDW [88]. However, in phase two, P3HB accumulation was induced by nitrogen limitation leading to a total biomass, %-P3HB CDW content and overall productivity of 150 g/L, 42% and 1.7 g/L/h, respectively [88]. 

Grousseau et al., conducted cultivations of *C. necator* DSM 545 on butyric acid to determine how maintaining continued cell growth would influence P3HB formation kinetics. These authors showed that NADPH formation via the Entner-Doudoroff pathway was enabled by anabolic demand. The result was a high carbon conversion efficiency where 0.89 mol-carbon in P3HB resulted from one-mole of carbon in the feedstock [89]. Indeed, this is an extraordinarily high carbon utilization efficiency. The total biomass, %-P3HB in the CDW and the overall productivity reached 46.7 g/L, 82% and 0.57 g/L/h, respectively [89]. 

Kim et al., reported the fed batch cultivation of *Methylobacterium organophilum* for P3HB production under potassium limitation. The methanol concentration was maintained at 2–3 g/L to avoid cell growth inhibition. P3HB accumulation accelerated when the concentration of potassium in the culture broth was reduced to less than 25 mg/L. The total biomass, %-P3HB CDW content and overall productivity reached 250 g/L, 52% and 1.86 g/L/h, respectively [90]. In other words, the fermentation produced 130 g/L P3HB [90]. 

*R. eutropha* ATCC 17699 was cultivated using fructose in stage one, and fructose/*γ*-butyrolactone during stage two under nitrogen limitation. Since *γ*-butyrolactone is metabolized into 4HB, the resulting product was P(3HB-*co*-4HB) [91]. To improve copolymer yields, cultivations were performed by fed-batch with DO-stat control for controlled feeding. Using the DO-stat strategy and at a 1.5:1 molar ratio of fructose to *γ*-butyrolactone, the total biomass, %-P(3HB-*co*-1.64 mol%4HB) in the CDW and overall productivity reached 48.5 g/L, 50.2% and 0.55 g/L/h, respectively [91]. 

## 4. PHA Production Using Genetically Modified Organisms

Non-PHA producing organisms can be genetically modified to biosynthesize PHAs (Figure 5). Furthermore, genetic modification of PHA and non-PHA producing strains provides a route to recombinant strains with improved kinetics for PHA production, a wider ability for substrate utilization (e.g., utilization of lignocellulose components), and changes in selectivity enabling the production of PHAs with unique structures. In addition, genetic engineering of strains has been used to improve substrate utilization from, for example, treated lignocellulose materials. In some cases, recombinant strain construction of, for example, *C. necator* and *E. coli* and are considered as strong candidate for commercial PHA production [92,93]. 

Recent strain engineering studies have focused on manipulating the metabolic flux by, for example, gene deletion or reducing gene expression for competing pathways that would redirect carbon away from PHA monomer formation [94]. Furthermore, genetic modification of production strains has been used to inhibit *β*-oxidation so that the PHA composition will more closely resemble that of the carbon source-fed [95]. A wide range of wild-type strains such as *A. hydrophila* 4AK4 [96], *M. extorquens* [97], and *Pseudomonas* [98,99] have been modified for enhanced copolymer yield, modification of PHA structure and incorporation of scl-and mcl-PHA monomers, respectively. 

*E. coli* has proved to be a valuable organism for genetic modification to attain highly productive PHA producing strains. Investigation of optimization of P3HB formation by *E. coli* has used tunable promotors to modulate expression levels of *pha**A*, *pha**B* and *pha**C* [95]. Another powerful tool is the ability to construct ribosomal binding site (RBS) libraries where the copy number of plasmids can be systematically varied [100]. Omission of enzymes such as the PHA depolymerase is valuable as the PHA production organism has no mechanism to carry out PHA hydrolysis, the reverse of PHA synthesis [101,102]. Table 4 provides representative examples of PHA production by recombinant *E. coli* and *C. necator* using HCDC methodologies.

*E. coli* possesses a rich genetic background and multiple available tools making it an ideal host for PHA biosynthesis [12]. The design and generation of recombinant *E. coli* strains has enabled the synthesis of a variety of PHAs. Furthermore, one can construct *E. coli* strains such that they can utilize a diverse set of feedstocks including carbon source mixtures derived from treatment of lignocellulose [103,104]. *E. coli* is capable of accumulating high contents of PHAs (80% to 90% of CDW) which enables large-scale PHA production. Also, Ren et al., claim there are economically viable method that can be applied to recover PHA from *E. coli* [105]. Due to the high accumulation of PHA, *E. coli* cells become fragile which enables efficient and easier product recovery [106]. Furthermore, produced PHA is not subjected to degradation during cultivations since it is normally constructed without an intracellular depolymerase enzyme [107]. 

Difficulties were encountered in identifying wild-type PHA producing strains that can use 4HB as the sole carbon source. To solve this problem, Le Meur et al., expressed a 4-HB-CoA transferase from *Clostridium kluyveri* orfZ in *E. coli*. In addition, the recombinant strain harbored the *PhaC* gene from *A. eutrophus*. By this approach, the developed strain gained the capability of producing P(4HB) during cultivations containing 4HB as the sole carbon source. That is, using glycerol for cell growth, 4HB for polymer formation, with exponential feeding to control the growth rate, and HCDC fed-batch operating conditions, P4HB formation reached a total biomass, %-P(4HB) in the CDW and overall productivity of 43.2 g/L, 33% and 0.21 g/L/h, respectively [50].

Ahn et al., expressed the *A. latus* genes *PhaA*, *PhaB*, and *PhaC* encoding in an *E. coli* strain. Interestingly, P3HB formation by this recombinant strain was more efficient than the corresponding recombinant *E. coli* strain was constructed using *PhaA*, *PhaB*, and *PhaC* from *R. eutropha*. The recombinant *E. coli* harboring the genes from *A. latus* was cultivated using a pH-state fed-batch culture and a 280 g/L sucrose-equivalent feed solution from concentrated whey. Whey is the liquid remaining after milk is curdled and strained. It is also a byproduct of cheese or casein manufacturing. The results are impressive as the total biomass, %-PHB in the CDW, overall productivity and carbon conversion efficiency reached 194 g/L, 87% (169 g/L P3HB), 4.6 g/L/h, and 0.45 g/g of P3HB per g of lactose [92]. These results are impressive and, if PHA copolymers could also be formed under similar conditions, this work provides guidance toward development of a commercially viable process.

The recombinant *R. eutropha* strain that expresses the *Rhodococcus aetherivorans* I24 PHA synthase gene and the hydratase gene (*phaJ*) from *P. aeruginosa* produced PHA containing HHx units [93]. The level of HHx in P(HB-*co*-HHx) was found to be a function of the acetoacetyl-CoA reductase activity. Cultivation of the recombinant *R. eutropha* strain on palm oil, under HCDC conditions, and by inducing nitrogen limiting conditions during PHA formation resulted in a total biomass, %-P(3HB-*co*-19 mol%3HHx) in the CDW and overall productivity of 139 g/L, 74% and 1.06 g/L/h, respectively.

Construction of the expression plasmids, pJRDTrcphaCABRe and pTrcphaCABRe was performed using the low and high copy number plasmid pJRDTrc1 and pTrc99a, respectively. Individually, these plasmids were expressed into *E. coli* XL1-Blue. The productivity of P3HB and biomass reached 2.8 g/L/h and 180 g/L, respectively, using glucose as the carbon source. The P3HB produced reached molecular weight values in the millions (3.5 × 10^6^ to 5.0 × 10^6^) and dispersity values remained low (~1.5) [108]. These ultrahigh molecular weight P3HB materials, like ultrahigh molecular weight polyethylene, provide advantaged mechanical properties.

The *E. coli* strain K24KL was constructed by deactivating the D-lactate synthesizing enzymes (ldhA) to produce P3HB from glycerol. This strain proved successful in increasing ethanol and P3HB production with a corresponding decrease in acetate formation. Analysis of the cofactor’s NADPH/NADP^+^ and NADH/NAD^+^ showed that *E. coli* K24KL possesses a higher ratio of the former (NADPH/NADP^+^). This led the authors to conclude that, the Idha mutation creates an intracellular environment with higher reducing capacity. By adopting fed-batch cultivation conditions, strain K24KL reached a total biomass, %-PHB in the CDW and overall productivity of 42.9 g/L, 63% and 0.45 g/L/h, respectively [109]. Insights from this work provide strategies for enhanced PHA formation from glycerol.

For improved plasmid stability, the kanamycin resistant gene was introduced into *E. coli* strain K1060. Cultivation of this strain was performed by a fed-batch operation in which the medium consisted of the agro-industrial by-products milk whey and corn-steep liquor. The total biomass, %-PHB in the CDW and overall productivity reached 70.1 g/L, 73% and 2.13 g/L/h, respectively [110]. 

Agus et al., expressed the *W. eutropha* PHA synthase (*PhaC*) in *E. coli* XL1-Blue cells. These workers than performed studies to assess how expression of PhaC effected P3HB production and molecular weight. IPTG (isopropyl-*β*-D-thiogalactopyranoside) functions as an inducer that controls the plasmid copies of PhaC. In other words, the concentration of IPTG in cultures provides a mechanism to control plasmid expression that results in PhaC formation. At 0.5 mM of IPTG that induces low PhaC expression, the recombinant *E. coli* strain reached a total biomass, %-PHB in the CDW and molecular weight of 178 g/L, 72% (128 g/L PHB), and 3 million g/mol, respectively [111]. In other words, the authors were successful in synthesizing PHB of ultra-high molecular weight.

The recombinant *E. coli* strain XL1-Blue which contained the *A. eutrophus* PHA synthase genes as well as the *E. coli* ftsZ gene that, in previous work, was found to increase P3HB formation efficiency by quelling filamentation, was used for P3HB formation. Cultivations were performed by fed-batch HCDC using glucose as the carbon source and thiamine for growth limitation. The total biomass, %-PHB in the CDW and overall productivity reached 156 g/L, 72% and 2.4 g/L/h, respectively [112]. 

The recombinant *E. coli* strain GCSC 6576 which contained the *R. eutropha* PHA synthesis genes as well as the *E. coli* ftsZ gene was studied to convert whey concentrate derived lactose to P3HB [113]. It consists of a 5% solution of lactose in water, with some minerals and lactalbumin (whey protein) [114]. Under fed-batch conditions, pH-stat control, and 210 g/L lactose equivalents from a concentrated whey solution, the total biomass, %-PHB in the CDW and overall productivity reached 87 g/L, 80% and 1.4 g/L/h, respectively [113]. These results highlight the potential to design of recombinant microorganisms for the efficient conversion of concentrated whey solution to PHB. 

Subsequently, Riedel et al., reported that cultivation of *R. eutropha* on low quality waste animal fats results in 45 g/L of biomass with 60% of PHA and a productivity of 0.4 g/L/h [115]. Sato et al., reported that recombinant *C. necator* H16 is capable of synthesizing high levels of P(3HB-*co*-19 mol%3HHx) using palm kernel oil and butyrate as carbon sources. Moreover, the authors showed that butyrate increased the 3-HHx fraction in phaA-deactivated mutant strains of KNK005 (AS). This strategy results in high biomass (171 g/L), HHx copolymer content in cells (81%) and, corresponding, high PHA titers (139 g/L) [116]. 

Povolo et al., developed a *C. necator* recombinant strain capable of using inexpensive carbon sources such as lactose and hydrolyzed whey directly from whey permeate with an enhanced PHA production capability. A contributing factor to increasing PHA productivity was eliminating the metabolic pathway for polymer degradation. The recombinant *C. necator* utilized hydrolyzed whey permeate (composed of glucose and galactose) as the sole carbon source such that the cells contain 30% PHB [117]. However, the authors did not mention the total biomass concentration, PHB yield and overall productivity. 

*P. putida* is a well-known producer of mcl-PHAs [118]. As discussed above, PHA production can be improved by deleting PHA depolymerase activity from the corresponding strain. Cai et al., constructed a recombinant *P. putida* KTMQ01 which accumulated 86% mcl-PHA of its CDW [119]. Le Meur et al., constructed a recombinant *P. putida* KT2440 strain to which the xylulokinase (XylB) and xylose isomerase (XylA) *E. coli* genes were inserted. The XylA and XylB genes help utilize the cost-effective substrate xylose, the main building block of the hemicellulose xylan, for mcl-PHA production. The resulting engineered *P. putida* KT2440 sequentially uptakes inexpensive carbon sources such as xylose and fatty acids (octanoic acid) for the cost-effective production of mcl-PHA. The cells reached 20% accumulation of mcl-PHA and authors did not determine final CDW and volumetric productivity [120].

Kahar et al., constructed a recombinant *R. eutropha* strain for high yield P(3HB-*co*-3HHx) production. *R. eutropha* PHA-negative mutant was built that harbored the phaC gene from *Aeromonas caviae*. Cultivations performed on soybean oil (20 g/L) by a fed batch process resulted in a 3HB copolymer consisting of 5 mol% 3HHx [121]. The total biomass, %-PHA in the CDW and overall productivity that reached 133 g/L, 72.5% and 1.0 g/L/h, respectively [121]. 

*Aeromonas hydrophila* 4AK4 produces P(3HB-*co*-3HHx) that contains 15 mol% HHx from dodecanoate [122]. To determine the factors that influence the incorporation of 3HHx in the copolymer, a recombinant *Aeromonas hydrophila* strain that expresses the genes *phaJ*, *phaC* and *phaP* from *Aeromonas punctate* were introduced individually or in combination. The authors discovered that expression of *phaC* alone enhanced the content of 3HHx in the copolymer from 14 to 22 mol% [122]. Co-expression of *phaC* with *phaP* and *phaJ* further increased the content of 3HHx in the copolymer to 34 mol%. The recombinant strain with *phaP* or *phaC* alone gave copolymer production in shake flask cultivations (48 h) that reached 4.4 g/L total biomass and 64% PHA in the CDW [122]. The authors concluded that by increasing the PHA synthase activity, higher contents of the 3HHx comonomer was incorporated whereas, by co-expressing *phaJ* with *phaP* or *phaC*, PHA production increased. Unfortunately, the authors did not assess whether high PHA production would be achieved in a fermenter using a HCDC protocol.

Towards the development of a strain that converts the unrelated carbon sources glucose and gluconates to P(3HB-*co*-3HHx), Qiu et al., built recombinant strains of *Pseudomonas putida* GPp104 and *Aeromonas hydrophila* 4AK4. This capability could eliminate the need for fatty acid substrates that can lead to foaming. The recombinant *A. hydrophila* 4AK4 expresses a cytosolic thioesterase-I, encoded by a truncated Tes A gene, to convert acyl-ACP into free fatty acids. Cultivation of the recombinant *A. hydrophila* 4AK4 strain on gluconate produced P(3HB-*co*-3HHx) containing 14 mol% HHx units [123]. Further genetic manipulations by overexpression of the P(3HB-*co*-3HHx) synthesis gene *phaPCJ* enlarged the copolymer content of 3HHx units to 19%. Moreover, these authors revealed that, recombinant *P. putida* GPp104, which harbors the *A. hydrophila phaC* gene that encodes the formation of 3HB/3HHx copolymers, *phaB* from *Wautersia eutropha* that encodes acetoacetyl-CoA reductase, and the *P. putida phaG* gene that encodes 3-hydroxyacyl-ACP-CoA transferase, resulted in PHA cell contents of 19% (*w*/*w*) with 5 mol% 3HHx units from glucose, a carbon source that is not related to HHx [123]. These results provide a roadmap to strategies that incorporate 3HHx units into PHA copolymers from unrelated carbon sources.

Ouyang et al., built a recombinant *A. hydrophila* 4AK4 strain that encoded the phbA and phbB from *R. eutropha* and *Vitreoscilla*, respectively. Cultivations of recombinant *A. hydrophila* 4AK4 were performed by a fed-batch process on the co-substrates dodecanoate and gluconate (1:1). Using dodecanoate only, the total biomass, %-P(3HB-*co*-12 mol%3HHx) content in the CDW and overall productivity reached 54 g/L, 52.7%, and 0.791 g/L/h, respectively [124]. In contrast, the wild-type strain produces total biomass, %-P(3HB-*co*-14.4 mol%3HHx) content and overall productivity of 40.4 g/L, 54.6% and 0.525 g/L/h, respectively [124].

The origin of sludge palm oil (SPO) is the palm oil milling industry. It is a solid that is generally considered difficult to use as a carbon source in cultivations. Budde et al., built a recombinant *C. necator* strain that encoded the phaC gene from *R. aetherivorans* I24 and phaJ gene from *P. aeruginosa* [125]. This engineered strain efficiently utilizes plant oils for P(HB-*co*-HHx) production. It was evaluated for its ability to utilize palm oil for P(3HB-*co*-3HHx) production [126]. To increase PHA productivity on SPO, fed-batch fermentations were conducted. The combination of the selected recombinant strain and cultivation conditions resulted in a total biomass, %-P(3HB-*co*-22 mol%3HHx) in the CDW and overall productivity of 88.3 g/L, 57% and 1.1 g/L/h, respectively [126].

A recombinant *E. coli* strain was built by encoding the *phaA*, *phaB* and *phaC* genes from *R. eutropha* PHA. This strain provided substantial benefits relative to the corresponding wild-type strain for PHA formation. Further improvements were realized by expressing the phaC gene from *A. latus* [127]. In 2002, Choi et al., used this strain for P(3HB-*co*-3HV) formation adopting a fed-batch feeding strategy and using glucose for biomass accumulation and the co-substrates propionic and oleic acids for PHA formation. They reported a total biomass, %-P(3HB-*co*-5.7 mol% 3HV) in the CDW and overall productivity of 42.2 g/L, 70% and 1.37 g/L/h, respectively [128]. 

Chen et al., constructed the recombinant *E. coli* strain by encoding the orfZ gene that expresses the *Clostridium kluyveri* 4HB-CoA transferase. The resulting strain, *Halomonas bluephagenesis* TD40, was evaluated for P(3HB-*co*-4HB) formation [129]. The use of this salt-tolerant strain allowed cultivations to be performed without taking precautions to maintain sterile conditions. HCDC of cultivations of *Halomonas bluephagenesis* TD40 were performed under fed-batch operating conditions, in 1 and 7 L fermenters for 48 h using glucose and *γ*-butyrolactone as carbon sources. The total biomass and %-PHA reached 70 g/L, and 63% of a P(3HB) copolymer containing 12 mol% 4HB [129]. Subsequently, this process was transferred to a 1000-L pilot scale fermenter which, by 48 h had a total biomass, %-PHA content and overall productivity of 83 g/L, 61% of the CDW that contained a P3HB copolymer with 16 mol-% 4HB and 1.04 g/L/h, respectively [129]. We conclude that *H. bluephagenesis* TD40 has excellent potential after further development to provide a platform for P(3HB-*co*-4HB) commercial production under open non-sterile conditions.

Poblete-Castro et al., constructed a recombinant *P. putida* KT2440 strain for PHA production on glucose by deletion of *gcd* (glucose dehydrogenase) and *gad* (gluconate dehydrogenase). The logic behind these deletions was to prevent gluconate and 2-ketogluconate formation. Fed-batch cultures were conducted under varying conditions and were used to assess mcl-PHA formation directly from glucose [130]. The first phase of biomass growth utilized exponential feeding with carbon limitation, whereas, for mcl-PHA formation in the second phase, substrate-pulse feeding, constant feeding and DO-stat feeding strategies were evaluated under nitrogen limiting conditions. The DO-stat feeding strategy gave the highest mcl-PHA formation such that the total biomass, %-mcl-PHA in the CDW and overall productivity reached 62 g/L, 67% and 0.83 g/L/h, respectively [130].

Yang et al., engineered *E. coli* strains to for PHAs containing aromatic repeat units from glucose, an unrelated carbon source [131]. For this purpose, the authors constructed a recombinant *E. coli* capable of producing D-phenyl lactate (PhLA). This involved the overexpression of isocaprenoyl-CoA:2-hydroxyisocaproate CoA-transferase as well as an engineered phaC from *Clostridium difficile*. The resulting recombinant *E. coli* was cultivated by a fed-batch process, using the co-substrates 3HB and glucose, such that the total biomass, %-P(38.1 mol% PhLA-*co*-61.9 mol% 3HB) in the CDW and overall productivity reached 25.27 g/L, 55% and 0.145 g/L/h, respectively [131]. Furthermore, the authors showed that other aromatic repeat units such as D-3-hydroxy-3-phenylpropionate and D-mandelate could be metabolized from glucose and incorporated into PHAs. This work highlights the ability to engineer PHA-producing strains that produce aromatic polyesters from unrelated renewable resources.

‘PHAomics’ highlights that a diverse range of PHAs, including those with block structures, can be formed by microbial PHA producers resulting in an expanded library of PHAs with unique properties [132]. The development of recombinant strains that enable the production of different types of random copolymer, homopolymers, block copolymers, and PHAs decorated with functional entities are displayed in Table 5. 

Engineered *P. putida*, *P. entomophila, P. mendocina, P. oleovorans, H. bluephagenesis and E. coli*, have been reported to produce homopolymers [133,134,135,136], scl-PHA random copolymer [137], scl- and mcl-PHA random copolymers [138,139,140,141], mcl-PHA random copolymers [133,134,142,143,144], block copolymers [100,107,133,134,141,145,146,147,148,149,150], functional PHAs [151,152,153,154], and PHA monomers [155,156].

*Pichia pastoris* has proved highly useful for the high-level expression of heterologous proteins [157]. Furthermore, *P. pastoris* has proved amenable to genetic manipulation. Also, this organism naturally synthesizes mcl-PHAs [158]. Vijayasankaran et al., reported that by the introduction of *R. eutropha* PHA biosynthesis genes, the recombinant *P. pastoris* strain showed an enhanced ability to accumulate P3HB [159]. This provides the opportunity to co-express the formation of PHAs and high-value proteins, an approach that can improve PHA economics.

A recombinant *E. coli* strain was built by encoding a phaC that can convert lactic acid-CoA to PHA repeat units. Furthermore, the strain was developed for the in vivo formation of lactic acid [160,161,162,163,164,165,166,167]. The resulting strain formed P(3HB-*co*-LA) with 4 to 47 mol% LA units. Changing the copolymer composition was accomplished by regulating the anaerobic culture conditions. The resulting P(3HB-*co*-LA) films were found to be comparatively pliable, flexible, and semi-transparent compared to both rigid homopolymers.

## 5. Enhancement of PHA Yield by *β*-Oxidation Inhibition

The mcl-PHAs usually occur as copolymers because the substrates used for biosynthesis are subjected to *β*-oxidation, resulting in the production of a mixture of repeat units that differ in chain length. In other words, even when the substrate is one structure, the resultant mcl-PHA is heterogeneous due to *β*-oxidation. To avoid *β*-oxidation of fatty acids, two methodologies have been developed; the first is to suppress or remove *β*-oxidation genes. Alternatively, inhibitors can be used to suppress enzymes catalyzing *β*-oxidation. Genetically modified organisms were found to be effective in obtaining a dominant repeat unit structure [168]. Jiang et al., also reported that cofeeding acrylic acid is a useful strategy to increase the direct incorporation of a selected substrate. In one example, a fed-batch cultivation of *P. putida* KT2440 on a substrate mixture consisting of glucose: nonanoic acid: acrylic acid (1: 1.25: 0.05, mass ratio) resulted in a total biomass, %-PHA of CDW content and productivity of 71.4 g/L, 75.5% (89 mol% 3HN) and 1.8 g/L/h, respectively [169]. In the absence of acrylic acid, the content of 3HN was reduced to 65 mol%. Also, *β*-oxidation-deleted mutants of *P. putida* or *P. entomophila* were effective in preparing mcl-PHAs that closely approached being homopolymers of a selected repeat unit [134,170].

Gao et al., deleted the *β*-oxidation gene of *P. putida* KT2440 towards synthesizing mcl-PHAs that closely approximated a homopolymer composition. In one example, cultivation of the *β*-oxidation deleted recombinant *P. putida* KT2440, using decanoic acid as the mcl-PHA producing substrate along with co-substrates glucose and acetic acid (2:8 g/g), the total biomass, %-PHA in the CDW and overall productivity reached 18 g/L, 59% and 0.32 g/L/h, respectively [171]. Remarkably, the mcl-PHA formed consisted of only 3HD units. In contrast, the wild strain cultivated under identical conditions, resulted in a total biomass, %-PHA content in the CDW and overall productivity of 39 g/L, 67% and 0.84 g/L/h, respectively [171]. Hence, the PHA yield was higher for the wild-type strain. However, the comonomer composition was 3HD: 3HO: 3HHx units in a molar ratio of 74:14:12.

Zhao et al., reported the construction of a recombinant *P. mendocina* by deletion of multiple genes associated with the *β*-oxidation pathway. The objective was to prepare mcl-PHA in higher yields that consisted of predominantly 3HD and 3HDD repeat units. Relative to the wild type strain, the recombinant *P. mendocina* had about a 5-fold increase in mcl-PHAs produced from sodium octanoate and sodium decanoate [142]. Using dodecanoic acid as the feedstock the mcl-PHA yield increased by 10-fold. The resulting mcl-PHAs have nearly uniform repeat unit structures that showed higher melting point transitions, mechanical and crystallization properties [142]. This approach demonstrated the potential of developing recombinant *P. mendocina* strains that both provide increased mcl-PHA production but also uniform compositions.

Oliveira et al., investigated mcl-PHA formation by wild-type and recombinant *β*-oxidation deleted strains of *P. putida* KT2440. Cultivations were conducted by a fed-batch operational mode using sugarcane biorefinery-derived hydrolyzed sucrose and decanoic acid as carbon sources. Using linear phase feeding strategy, *P. putida* KT2440 reached a total biomass, %-PHA content in the CDW and overall productivity of 53.4 g/L, 33% and 0.4 g/L/h, respectively [172]. The composition of the mcl-PHA formed consisted of C10:C8:C6 repeat units in a molar ratio of 84:14:2. However, using the same cultivation conditions, the *β*-oxidation deleted *P. putida* strain reached a total biomass, %-PHA of CDW and overall productivity of 24.6 g/L, 42% and 0.25 g/L/h, respectively with C10:C8:C6 mol% composition of 95:5:0 [172]. Hence, while the *β*-oxidation knockout mutant had lower productivity, it provided a means to produce an mcl-PHA that closely approaches a 3HD homopolymer.

A *β*-oxidation pathway modified mutant of *R. eutropha* was explored for P(3HB-*co*-3-HHx) production from soybean oil [173]. Deletion of fadB1 (enoyl-CoA hydratase/3HA-CoA dehydrogenase) in recombinant *R. eutropha* strains with other genes encoding (R)-enoyl-CoA hydratases, a 6–21% increase in 3-HHx content in the copolymer was observed [173]. This was attributed to an increased availability of mcl-2-enoyl-CoAs by partial impairment of *β*-oxidation. This work provides a useful strategy to increase the 4HHx content in copolyesters synthesized from fatty acids.

## 6. PHA Production Using Mixed Cultures

Mixed microbial consortia (MMC) provide advantageous routes for PHA production as cultivations are conducted in open systems, circumventing the need to maintain sterility, which can reduce operating costs [174,175,176,177,178,179] (Figure 6). As will be further elaborated below, agro-industrial and municipal waste streams offer low-cost substrates and sources of PHA producing microbes. Volatile fatty acids (VFAs) are a valuable PHA substrate produced by acidogenic fermentation during anaerobic digestion. Afterwards, activated (microbe containing) waste sludge generally undergoes a series of feast/famine feed cycles (visualized within the dashed border of Figure 6) to maximize the population of PHA-producing microbes. Other conditions, such as pH, cultivation time, and temperature are modified to further maximize PHA production. Lastly, the maximum PHA-producing MMC is utilized in subsequent PHA fermentation processes.

Harnessing MMCs for PHA production requires a first step, often carried out in sequential batch cultivations where selection and enrichment occur resulting in a microbial consortium with a high capacity for PHA formation. This normally required the application of transient conditions. Then, cultivation conditions are applied that further maximizes PHA formation [180,181].

MMC technology often utilizes volatile fatty acids, VFAs (i.e., acetic, propionic, butyric, and valeric acids), as substrates for PHA production [182,183,184]. These substrates are available via anaerobic digestion processes of wet wastes such as that from biomass fractions, foods, wastewater sludge, fats, animal waste, and more. VFA are advantageous over glucose and other carbohydrate feed sources since they are highly oxidized providing relatively higher equivalent carbon and energy [185]. The relative composition of VFA will determine the PHA composition as, for example, high propionic acid contents will be metabolized to 3HV units whereas even-carbon chain length VFAs will likely form P3HB [186,187].

The diversity of microorganisms in MMCs provides multiple PHA production pathways. A batch reactor that is acetate-fed run with a biomass residence time of one day, and 12 h cycles of feast-famine (F-F) conditions, gave an MMC enriched in PHA producers [180]. Subsequently, fed-batch cultivation experiment using the MMC mixed culture under growth limiting conditions resulted in high PHA storage cell accumulation (89% of the CDW) and high P(3HB) production rates (~1.2 g/g/h) [180]. The dominant microorganism in the cultivation was a *Gammaproteobacterium* that proved to have low similarity to known bacteria.

Jiang et al., used lactate as well as lactate/acetate mixtures to obtain PHB producing MMC’s and investigated the use of lactate and a mixture of lactate and acetate for enrichment of PHB producing mixed cultures [188]. Operational conditions that previously resulted in a performing strain for PHB production from acetate was adopted for this work [189]. The enrichments from acetate and lactate used microbes from activated sludge. As above, the dominant microorganism from the enrichment on lactate was a *Gammaproteobacterium* that reached in 6 h PHB contents in its CDW of 90% [189]. When an acetate/lactate mixture was used during the enrichment, the dominant microorganisms were the bacteria *Thauera selenatis* and *Plasticicumulans acidivorans* that reached PHB contents in the CDW of 84% in 8 h [189]. Previous work with strains *Thauera selenatis* and *Plasticicumulans acidivorans* showed that they were capable of high PHB accumulation.

Cui et al., performed enrichments to obtain PHA forming MMCs using starch, glucose and acetate as substrates. Aerobic extended-time dynamic feeding intervals were used. The organisms were exposed to long-term aerobic dynamic feeding periods. Following 350 enrichment cycle intervals under F-F regimes, the MMCs enriched by feeding starch, glucose and acetate accumulated 27%, 61% and 65% PHA in their CDWs, respectively [190]. Sequencing studies revealed that, in addition to PHA forming bacteria, microbes that are non-PHA producing also survived. Using acetate for enrichment, the dominant PHA forming genera were *Stappia* and *Pseudomonas*. In contrast, enrichments conducted using glucose resulted in the PHA forming genera *Vibrio*, *Piscicoccus* and *Oceanicella*. For enrichments on starch, the sole PHA forming Genus was *Vibrio*.

Palmeiro-Sánchez et al., explored how imposing recurrent sodium chloride concentrations (NaOH, 0.8 g Na^+^/L) would influence the resulting MMC cultures ability to produce PHA. Enrichments were on a mixture of C2-C5 aliphatic VFAs where the major constituents were acetic and propionic acids (54 and 27 mol%, respectively) [191]. PHA production by the MMC reached 53% of the CDW and the corresponding copolymer composition was P(3HB-*co*-27 mol%3HV) [191]. A comparative study where the NaOH was not imposed during enrichments resulted in a relatively lower ability of the resulting MMC to produce PHA.

Cavaillé et al., (2016) developed an MMC that consisted of PHA forming microbes from activated sludge. Cultivations were performed under non-sterile conditions using acetic acid as the carbon sources under phosphorous limitation. A stable continuous process for PHA formation was reached where the CDW of effluent cells reached 74% [192].

Thermophilic and thermotolerant strains can be used to develop MMC’s for PHA formation to reduce essential heating and cooling operations that are energy intense processes [193]. For such cultivations, adequate oxygen must be accessible to microbes at working temperatures. Benefits of working with thermotolerant MMC enrichments include faster diffusion, higher solubility of substrates, higher PHA production rates and a lower risk of culture contamination [194].

Toward reducing the costs of PHA formation, MCCs were carried out using low cost agro-industrial wastes that include paper mill effluents, cannery effluents, municipal sludge, saponified sunflower oil, fermented sugar cane molasses, industrial and domestic wastewaters and oil mill effluents [195]. Johnson et al., studied MMCs with high storage capacity for PHA production. Acetate was fed for 24 h in the batch reactor to accumulate biomass with F-F cycles of 12 h. This strategy results in accumulation of 89% PHA in the CDW in 7.6 h with an overall PHA productivity of 1.2 g/g/h [196].

Lorini et al., reported on application of sequencing batch reactors for cultivation of mixed culture with uncoupled carbon and nitrogen feeding. The different organic load rates studied (4.25 to 12.72 g COD/L·d) results in PHA productivity of 0.1 g PHA/L/h at optimum conditions [197]. Silva et al., explored how nitrogen feeding would influence PHA formation by MMCs. A mixture of C2 and C3 VFAs were used as carbon sources at an organic load of 8.5 g COD/L·d with concurrent feeding of (NH_4_)_2_SO_4_. This strategy resulted in the accumulation of PHA with up to 20 ± 1% *w*/*w* HV) [198]. Liu et al., reported mcl-PHA formation by *Pseudomonas-Saccharomyces*, using xylose as carbon source, results in 152.3 mg/L PHA [199]. The presence of *S. cerevisiae* in the consortium improved the sedimentation of cell mass.

## 7. Industrial/Agro-Industrial Waste for Production of scl-and mcl-PHA

Despite the promising properties of PHAs, they remain uncompetitive with conventional plastics due to both high cost and performance shortfalls. Significant improvements in performance can be realized by polymer formulation during processing [200,201,202], but this is beyond the scope of this review. The cost of carbon substrates constitutes ~50% of the total manufacturing cost [203]. Therefore, a major research effort is underway focused on reducing the production cost by utilizing low or no-cost waste materials as carbon sources [20,204,205].

The yield and monomer composition during PHA formation varies depending on the type of microbial strain and carbon sources. The use of commercial sugars in PHA production by bacterial fermentation leads to high-cost processes. Therefore, to achieve cost effective PHA production, in addition to reducing energy utilization and efficient downstream processes, it is also critical to move to low-no cost feedstocks such as cellulose, lignocelluloses, hemicelluloses and sugars derived therefrom. The sustainable sources of feedstock for PHA production are depicted in Figure 7.

In addition to the above, large waste quantities are generated from numerous sources such as agriculture (including lignocellulosic materials), food, industrial and municipalities. An opportunity exists to create value from these zero cost feedstocks that would otherwise be landfilled. Wastes may be enriched in valuable carbon sources for PHA production such as carbohydrates of various complexity and fatty acids. Pretreatments that may involve biochemical, chemical or physical processes can significantly enhance the contents of wastes to more readily metabolizable substrates. The representative literature on HCDC using renewable feedstock are presented in Table 6.

### 7.1. Lignocellulosic Feedstock

A great opportunity exists to exploit the estimated 200 billion tons global production of lignocellulosic feedstocks [206,207]. It is well known that efficient utilization of lignocellulosic feedstocks comes with numerous challenges since these materials were designed by nature to be difficult to access by microbes to enable their long lifetime as plant structural materials.

Furthermore, phenolic compounds generated by lignin can inhibit PHA producing microbial growth and metabolic processes [208]. Fortunately, due to the numerous products envisioned by conversions of highly abundant lignocellulose by integrated biorefinery processes, there have been continuous improvements in pretreatment processes that use, for example, physicochemical, alkaline and enzyme-mediated reactions to achieve high sugar yields for production of biofuels and other valuable products [209]. For example, xylose is highly abundant in hemicellulose that can be liberated by efficient enzymatic or chemical processes for use as primary feedstocks for scl-PHA formation by *P. cepacia* ATCC 17759 [210].

About 25% *w*/*w* of wheat straw (WSH) consists of hemicellulose that are rich (80 dry wt) in pentoses [211]. Cesario et al., reported fed batch cultivations of *B. sacchari* DSM 17165 for PHA production from wheat straw hydrolysate that provides primarily arabinose, xylose and glucose [211]. *B. sacchari* was shown to efficiently metabolize these sugars. To promote polymer accumulation, KH_2_PO_4_ limitation was imposed by reducing its initial concentration to 3 g/L. Fed-batch *B. sacchari* cultivations did not exhibit catabolite repression that can occur when fed wheat straw hydrolysate sugars and reached a total biomass, %-PHB in the CDW and an overall productivity of 146 g/L, 72% and 1.6 g/L/h, respectively [211].

Cesario et al., adopted a fed-batch feeding strategy for the formation of P(3HB-*co*-4HB) by *B. sacchari* [212]. These authors used as carbon source a xylose-rich hydrolysate of lignocellulose and, *γ*-butyrolactone served as a 4HB unit precursor. This strategy results in a total biomass, %-P(3HB-*co*-6 mol%4HB) in the CDW and an overall productivity of 88 g/L, 27% P(3HB-*co*-6 mol%4HB) and 0.5 g/L/h, respectively [212]. These authors claimed that this is the first demonstration where *B. sacchari*, a strain that produces PHA from xylose-rich lignocellulosic hydrolysates, formed P(3HB-*co*-4HB) by using as co-substrate *γ*-butyrolactone. Thus, to effectively produce PHA from hemicellulose, the corresponding microbes must have the metabolic machinery to utilized hemicellulose derived pentoses and hexoses. The composition of wheat straw hydrolysate is 35.5% of glucose, 24.7% of xylose, 3.3% of arabinose, 16.4% of lignin, 8.8% of ash and 1.7% of raw proteins [213].

A promising feedstock from lignocellulose or derived sugars is levulinic acid (4-oxovaleric acid) as it can be produced with increasing efficiency from lignocellulose and various waste materials [214]. Levulinic acid has 5-carbons and, as such, is a good precursor for formation of 3HV units in PHAs. While it can serve as a primary substrate for cell growth of some PHA producing organisms, it may also cause inhibitory effects such that its concentrations in cultivations need to be kept below 2 g/L [215,216].

Xu et al., used a media consisting of wheat hydrolysate. By fed-batch cultivation of *W. eutropha* they reached a total biomass, %-PHB in the CDW and an overall productivity of 175 g/L, 93% PHB (i.e., 162.8 g/L) and 0.88 g/L/h, respectively [217].

Sathiyanarayanan et al., explored the sponge-associated endosymbiont *B. subtilis* MSBN17 for PHB accumulation from pulp industry waste (PIW) as the major substrate and tamarind kernel flour as a co-substrate. PIW was provided to cultures at 12, 28 and 40 h time intervals. Under optimized conditions, the total biomass, %-PHB in the CDW and the overall productivity of *B. subtilis* MSBN17 was 24.2 g/L, 51.6% and 0.26 g/L/h, respectively [218].

Dietrich et al., reported a HCDC of *Paraburkholderia sacchari* IPT 101 LMG 19450 for PHB production from hard wood hydrolysate as the sole carbon source. These authors compared a synthetic hydrolysate that consisted of equivalent substrates (acetate, xylose, glucose) with a wood hydrolysate. PHB formation was performed during 52 h HCDCs [219]. Interestingly, the PHB concentration (22 g/L) was relatively lower using the synthetic hydrolysate medium. This can be explained by the presence of additional nutrients in wood hydrolysate and the corresponding specific growth rates that were higher on the wood hydrolysate (0.36 vs. 0.33 h^−1^). The cultivation on wood hydrolysate, the total biomass, %-PHB in the CDW and an overall productivity were 59.5 g/L, 58% and 0.72 g/L/h, respectively [219]. In comparison, the synthetic hydrolysate resulted in %-PHB in the CDW and an overall productivity of 55% and 0.46 g/L/h, respectively [219]. This report on the conversion of hard wood hydrolysate is promising but, as in the above work by Sathiyanarayanan et al. on PIW, further process and potentially cell engineering will be needed to achieve industrially relevant product titers and productivity.

### 7.2. Waste Glycerol

Crude glycerol from biodiesel production that, annually, is about 150 million gallons, generates 50 million kg of crude glycerol [220]. Problems encountered with the use of crude glycerol are co-generated impurities (i.e., fatty acids, salts, methanol) that lower its value. With continuous improvements in triglyceride production by oleaginous yeasts from lignocellulosic sugars, it is anticipated that triglyceride production from non-food sources will be available in increasing amounts as will crude glycerol [221]. This has spurred research to utilize crude glycerol as a carbon source for the biotechnological production citric acid, 2,3-butane diol, PHB, 1,3-propane diol and more [222,223,224].

Cavalheiro et al., investigated the conversion of crude glycerol to P3HB copolymers with 4HB and 3HV repeat units [23]. As discussed above, incorporating *γ*-butyrolactone as a co-feedstock will lead to the formation of 4HB units, whereas the co-substrate propionic acid can be metabolized to both 3HV and 4HB units. Incorporation of 4HB monomers was promoted by *γ*-butyrolactone. After optimizing dissolved oxygen in cultivation media, using crude glycerol and *γ*-butyrolactone as cosubstrates in cultivations of *C. necator* DSM 545 resulted in a total biomass, %-PHA in the CDW and overall productivity of 30.2 g/L, 36.1% P(3HB-17.6 mol%4HB) and 0.17 g/L/h, respectively [23]. In contrast, by using crude glycerol, *γ*-butyrolactone and propionic acid as co-feedstocks, the total biomass, %-PHA in the CDW and overall productivity reached 45.25 g/L, 36.9% of P(3HB-43.6 mol%4HB-6 mol%3HV) and 0.25 g/L/h, respectively [23]. Production of PHA from crude glycerol helps reduce production costs with concomitant glycerol valorization. Using the same production strain, Cavalheiro et al., studied PHB production from crude glycerol during fed-batch cultivations of *C. necator* DSM 545. They reported a total biomass, %-PHB in the CDW and an overall productivity of 68.8 g/L, 50% and 1.1 g/L/h, respectively [225]. 

Mozumder et al., reported a three-stage control strategy that was claimed to be organic substrate-independent for automated substrate feeding in a two-phase fed-batch culture [41]. This sensitive and robust feeding strategy was applied to *C. necator* DSM 545 cultivations for conversion of crude glycerol to PHB at experimentally determined optimal substrate concentrations. To reach maximal cell biomass, the feeding strategy combined exponential feeding and alkali-addition monitoring. Subsequently, the substrate feeding rate was kept constant while nitrogen feeding was stopped, resulting in PHB accumulation. This resulted in a total biomass, %-PHB in the CDW and overall productivity of 104.7 g/L and 62.7% and 1.36 g/L/h [41]. This work is certainly a promising example where high PHB titers (66 g/L) were achieved using crude glycerol as the sole carbon source.

Salakkam and Webb investigated whether biodiesel production by-products rapeseed meal and crude glycerol could function synergistically for PHB production by *C. necator* DSM 4058. Rapeseed meal processing provides a media component rich in free amino nitrogen. Adopting a fed-batch cultivation strategy for the co-substrates crude glycerol and processed rapeseed meal, without further nutrient supplements, resulted in a total biomass, %-P3HB in the CDW and overall productivity of 28.9 g/L, 85.8% and 0.21 g/L/h, respectively [226].

A strategy to improve PHA production economics is to use or develop production strains that also generate high value coproducts. Kumar et al., used glycerol as a carbon source for cultivations of *Paracoccus* sp. LL1 that produced carotenoids along with PHA. An enhancement of total biomass was achieved through cell retention, resulting in 24.2 g/L biomass that consisted of 39.3% PHA and 7.14 mg/L carotenoids [227]. Volova et al., reported a fed-batch cultivation of *C. eutrophus* B-10646 for PHB production from glycerol of different purification degrees (99.5%, 99.7% and 82.1% purity) in 30 L and 150 L fermenters. When glycerol of purity 99.3% was used in a 30 L fermenter, the maximum total biomass concentration and PHB content in the CDW was 69.3 g/L and 72.4%, respectively [228]. A further increase in glycerol purity to 99.7% resulted in similar results at 30 L. Interestingly, the use of crude glycerol of purity 82.1% did not significantly decrease total biomass concentration (69.3 g/L) or PHB content in the CDW (78.1%). When PHB biosynthesis by *C. eutrophus* B-10646 on glycerol of purity 99.7% was scaled-up from 30 L runs to 150 L, the total biomass, %-PHB of the CDW and overall productivity reached 110 g/L, 78% and 1.83 g/L/h, respectively [228]. Given the promising results at 30 L with crude glycerol, it would be interesting to see if similar improvements in PHB production efficiency from crude glycerol would be obtained using the same process conditions at 150 L. Zhu et al., reported on cultivation of *B. cepacia* ATCC 17759 for biosynthesis of PHB using crude glycerol. The fermentation process, when scaled to 200 L, gave 23.6 g/L biomass of which 31% was P3HB [229].

Kachrimanidou et al., reported cultivations of *C. necator* DSM 7237 operated in fed batch mode that utilized sunflower meal and crude glycerol as co-substrates for P3HB formation. When levulinic acid and sunflower meal hydrolysate were used as co-substrates, the product form was P(3HB-*co*-3HV) [230]. Sunflower meal served as a source of both inorganic phosphorus and free amino nitrogen that were critical components to achieve substantial PHA formation and cell growth. Cultivations conducted in fed-batch mode resulted in a total biomass, %-PHB of the CDW and overall productivity of 37 g/L, 72.9% and 0.28 g/L/h, respectively [230]. Continuous feeding of levulinic acid results in a total biomass, %-PHB of the CDW and overall productivity of 35.2 g/L, 66.4% P(3HB-*co*-22.5 mol%3HV) content and 0.24 g/L/h [230]. This further reinforces the role of levulinic acid as a precursor substrate for 3HV formation and incorporation in PHAs.

### 7.3. Sugar-Cane Molasses as Carbon Source

Molasses is a sugar-rich byproduct from sugar beet and sugarcane refining into sugar [231]. It has been extensively studied as a substrate for industrial-scale fermentation processes due to its abundance and low price. Molasses contains predominantly 50% sucrose with lower quantities of glucose and fructose, about 4% protein, trace elements calcium, magnesium, potassium, and iron and vitamins H or B7 [232]. Jiang et al., used sugarcane molasses as substrates for PHA formation by *P. fluorescens* A2a5. In a 5 L bioreactor, the total biomass, %-P3HB in the CDW and overall productivity reached 32 g/L, 68.75% and 0.23 g/L/h, respectively [233].

Kulpreecha et al., reported the use of urea and sugarcane molasses as nitrogen and carbon sources, respectively, for cultivations of *B. megaterium* BA-019. Cultivations were conducted in fed-batch mode under pH-stat feeding control. Under conditions where the molasses feeding solution was 400 g/L and the C/N molar ratio was 10:1, the total biomass, %-PHB in the CDW and overall productivity reached 72.6 g/L, 42% and 1.27 g/L/h, respectively [38]. In a later study, Kanjanachumpol et al., used the same production organism and substrates but changed the following process variables: C/N ratio was increased to 12.5/1 and an intermittent feeding strategy was implemented [234]. As a result, the total biomass, %-PHB in the CDW and overall productivity reached 90.7 g/L, 45.8% and 1.73 g/L/h, respectively [234].

Sugarcane vinasse is the final by-product of biomass distillation from ethanol production using substrates such as sugar crops (beet and sugarcane). It consists of primarily acid-insoluble nitrogen [235], and is rich in melanoidins and phenolics [236]. Dalsasso et al., reported the fed-batch cultivation of *C. necator* for PHB production using sugarcane vinasse and molasses as mixed substrates. Addition of vinasse to the molasses cultivation medium resulted in an increase from 0.19 to 0.36 h ^−1^ in the maximum specific growth rate [237]. This was attributed to the presence of organics in vinasse that were rapidly consumed. Molasses was added at two-time intervals and the cultivation gave 20.9 g/L biomass of which 56% of the CDW is PHB [237]. The nitrogen content in both molasses and vinasse (around 1.8 g/L) caused an uncharacteristic rise during the P3HB formation stage in the biomass concentration.

### 7.4. Green Grass as Carbon Source

Recently, researchers investigated the potential of transforming green grass into substrates for the formation of mcl-PHA production. Davis et al., reported on mcl-PHA formation from perennial ryegrass biomass. The pretreatment of grass biomass results in highly digestible (~75%) substrate rich in C5 and C6 sugars for mcl-PHA formation. While the authors provided information on the fact that *P. putida* W619 and *P. fluorescens* 555 utilized the pretreated perennial ryegrass substrate forming 25-34% mcl-PHA of the CDW, they did not report the total biomass and overall productivity [238].

### 7.5. Starch as Carbon Source

Many researchers have explored starch as low-cost carbon source in industrial fermentation processes. While starch is not a waste source as it can be directly used by humans in foods; the high productivity of corn by US farmers has created excess starch that is available for fermentation to produce chemicals. As the demands grow for starch in fermentation to produce chemicals or the population increases, it will be necessary to avoid further consumption of starch that is needed to feed the population. That said, some bacterial strains do not produce *α*-amylase; hence this enzyme needs to be added externally to hydrolyze starch. Haas et al., reported fed-batch HCDC of *R. eutropha* NCIMB 11599 using saccharified (converted to sugars) waste potato starch as the carbon source. They reported a total biomass, %-PHB in the CDW and overall productivity of 179 g/L, 52.5% and 1.47 g/L/h, respectively [239]. While residual maltose accumulated in the bioreactor, it did not cause a noticeable inhibition of cell growth or metabolic processes leading to PHB. Chen et al., explored extruded starch for production of PHA using extremely halophilic Archaeon *H. mediterranei*. Starch was extruded with yeast extract at a weight ratio of 1/1.7 to attain a favorable carbon-to-nitrogen ratio in cultivations. Fermentation were operated in fed-batch mode using pH-stat control. The resulting total biomass, %-PHA in the CDW and overall productivity reached 39.4 g/L, 50.8% P(3HB-*co*-10.4 mol%3HV) and 0.29 g/L/h [240].

### 7.6. Whey, Wheat, and Rice Bran as Carbon Sources

Under the section of PHA production using genetically modified organisms (Section 3), we presented work on whey utilization by recombinant *E. coli* [92,241]. Park et al., used the recombinant *E. coli* strain CGSC 4401 that harbors biosynthetic PHA biosynthetic genes of *A. latus* [242]. In contrast to the above work by Ahn et al. [92], they conducted studies at low lactose feed rates and did not use concentrated oxygen for oxygen supply to cultures. By maintaining lactose concentrations (<2 g/L), low cell growth and PHA contents resulted (12 g/L and 9%, respectively) [237]. However, at 20 g/L lactose, the total biomass concentration, %-PHB in the CDW, and overall productivity reached 51 g/L, 70%, and 1.35 g/L/h, respectively [242]. This fermentation, when scaled to 300 L, resulted in a total biomass, %-PHB in the CDW and overall productivity of 30 g/L, 67% and 1 g/L/h, respectively [242].

Wheat bran, that contains substantial quantities of hemicellulose and cellulose, is a rich source of proteins, carbohydrates, and other minerals. Annamalai and Sivakumar used *R. eutropha* NCIMB 11599 as the P3HB production strain and wheat bran hydrolysate as the source of carbon and other nutrients. This work resulted in a total biomass, %-PHB in the CDW and overall productivity of 24.5 g/L, 62.5%, and 0.255 g/L/h, respectively [243]. An important observation by these workers is that wheat bran hydrolysate did not cause any apparent toxicity to the PHB production strain.

Huang et al., reported the production of PHA by *H. mediterranei* using as carbon sources corn starch and rice bran that was first extruded to enhance the ease by which cells can utilize these substrates [244]. This strain functions in hyper-saline conditions that virtually circumvents contamination problems. Furthermore, PHA downstream processing is simplified as the haloarchaea is readily lysed in distilled water giving PHA pellets that can be recovered by centrifugation at low speeds. Rice bran and starch, extruded in a 1:8 (g/g) ratio, was used as the fermentations source of carbon nutrient. The authors employed a pH-stat control strategy to maintain pH at 6.9 to 7.1 in a 5 L fermenter. Cultivations were operated in repeated fed-batch mode and the pH was maintained between 6.9–7.1 using a pH-stat. They reported a total biomass and %-PHB in the CDW of 140 g/L and 55.6% [244]. The development of processes for PHA formation by halophiles such as *H. mediterranei* and increasing rice bran and starch access by cells using extrusion are notable achievements by the authors.

### 7.7. Waste Vegetable Oils and Plant Oils as Carbon Sources

Triglycerides are important substrates for PHA production as they have a high density of carbon by weight, are directly metabolized to acetate (an excellent PHA substrate), are used to form 3HHx units in copolymers and can form mcl-PHAs. However, triglyceride derived fatty acids produced from edible oils is not economically feasible. Also, the widespread use of edible oils as fermentation feedstocks, oleochemicals and biodiesel production can lead to a food crisis [245]. Waste frying oil has been reported as a useful substrate by many bacterial strains for PHA formation. In cases where triglycerides are used directly, their rate of hydrolysis to fatty acids and glycerol may limit cell growth and product formation. Approaches to mediate these drawbacks are: (i) conversion of triglycerides to fatty acids or their methyl esters prior to cultivations, (ii) amending media with lipases and (iii) emulsification of triglycerides with non-toxic surfactant to increase their availability [246].

Tufail et al., investigated the utility of waste frying oil as a substrate for PHA formation. Cultivations were conducted in shake flasks containing 2% waste frying oil. Oil droplets were formed by sonication and subsequent shaking. The maximum PHA formed from waste frying oil (53.2% PHA and total biomass 23.7 g/L) was by *P. aeruginosa* (KF270353), cultured for 72 h at 100 rpm [247].

Obruca et al., investigated PHA formation by *C. necator* H16 using waste frying rapeseed oil as the carbon source. Cultivations operated in fed-batch mode resulted in a total biomass, %-PHA in the CDW and overall productivity of 138 g/L, 76% (i.e., 105 g/L PHB) and 1.46 g/L/h, respectively [248]. Furthermore, the substrate conversion efficiency reached 0.83 g PHB per g oil. This is an extraordinary result showing the value that waste rapeseed oil can provide in the cultivation of prepared PHAs. By addition to cultivations of 1% *v/v* propanol, biomass and PHA formation was increased. Also, the PHA formed contained 8 mol% 3HV units in the copolymer structure [248].

Cruz et al., focused on processes that will lead to efficient conversions by *C. necator* DSM 428 of waste cooking oil (WCO) to PHA. Operational strategies to increase cultivation efficiencies included exponential substrate feeding and use of DO-stat for process control. Utilizing exponential feeding, the total biomass, %-PHB content in the CDW and overall productivity reached 21.3 g/L, 84% and 0.1875 g/L/h, respectively [249]. A substantial improvement in the productivity of PHB formation (to 0.525 g/L/h) resulted from controlling pH with ammonium hydroxide as well as using a DO-stat for process regulation [249].

Ruiz et al., used *P. putida* KT2440 as the PHA production strain and fatty acids derived from WCO as the carbon source. The implemented feed strategy delayed the stationary phase to enhance biomass and PHA formation. Use of intermittent feeding and conditions that led to HCDC, *P. putida* KT2440 reached a total biomass, %-mcl-PHA content in the CDW and overall productivity of 159.4 g/L, 36.4% (i.e., 57 g/L mcl-PHA) and 1.93 g/L/h, respectively [250]. This bioprocess provides a strong foundation that, with further process optimization, could lead to a commercially viable process for conversion of WCO to PHA.

Ruiz et al., using *P. chlororaphis* 555 as the production strain, developed a HCDC bioreactor-based process to convert WCOs from restaurants to mcl-PHA. The composition of the WCO is palmitic acid (7.9%), stearic acid (42.3%), oleic acid (42.3%), linoleic acid (32.2%), and others (0.7%) [251]. In batch bioreactor experiments, when 60 g/L of WCO was supplied during a 30 h fermentation, the total biomass, %-mcl-PHA content in the CDW and overall productivity reached 45.5 g/L, 19.8%, and 0.30 g/L/h [251]. The authors then implemented a pulse feeding strategy controlled by the DO-stat. That is, increase in the DO above 20% prompted substrate feeding. They also implemented nutrient limitation (phosphorous) during PHA formation. The resulting cultivation in which 140 g WCO/L was supplied over 48 h resulted in a total biomass, %-PHA content, and overall productivity of 73 g/L, 19%-P(20 mol%3HDD-37 mol%3HD-36 mol%3HO-7 mol%3HHx) and 0.29 g/L/h, respectively [251].

Thuoc et al., reported the use of glycerol and waste fish oil as substrates for PHA formation. The PHA-forming bacterium *Salinivibrio* sp. M318 was isolated from fermenting shrimp paste. By operating cultivations in fed batch mode, the total biomass, %-PHB and overall volumetric productivity reached 69.1 g/L, 51.5% and 0.46 g/L/h, respectively [252]. The isolation and development of PHA-forming microbes that can use waste materials from aquaculture is important given the large volume of these waste materials.

Da Cruz Pradella et al., explored pulsed feeding of soybean oil to attain high yield and productivity of PHB in *C. necator* DSM 545. By using pulsed feeding strategy, the total biomass, %-PHB and overall volumetric productivity reached 83 g/L, 80% and 2.5 g/L/h, respectively [253]. Shang et al., reported mass production of mcl-PHA from corn oil hydrolysate in *Pseudomonas putida* KT2442. By using fed-batch cultivation, the total biomass, %-PHB and overall volumetric productivity reached 103 g/L, 27.1% and 0.61 g/L/h, respectively [254]. 

### 7.8. Wastewater for PHA Production

Wastewater is a potential source of carbon and other nutrients for PHA production. Ryu et al., focused on swine wastes water augmented with yeast extract, inorganic salts and glucose as substrates for *A. vinelandii* UWD cultivations. Supplementing batch cultures of swine wastewater with 30 g/L glucose, resulted in about a six-fold increase in both cell growth and PHA formation [255]. These increases varied based on the extent of swine wastewater dilution. At 50% dilution, the total biomass reached 9.4 g/L that contained 58%-PHA [255]. This work is an example of where, to achieve suitable production of PHA from a waste source, supplementation with another carbon source is required.

Yan et al., assessed the potential use of activated sludge as a source of carbon and nutrients for PHA formation. Substrates were obtained from full-scale wastewater plants that include those from cheese, starch, municipal and pulp/paper manufacturing facilities. These wastewaters were used as sources of PHA producing microbes for shake flask experiments. VFAs from wastewaters were believed to function as carbon substrates. Activated sludge from pulp and paper mills gave maximal PHA formation (43% of suspended solid dry weight) [256].

## 8. Global PHA Producer Companies at Pilot and Industrial Scale

Pilot- and industrial-scale PHA manufacturers along with product trade names, PHA type, raw material, production capacity and estimated prices are listed in Table 7. At present, numerous companies are pursuing the manufacture of PHB and its copolymers at pilot and industrial scales. Some PHA-producing companies are focused on PHA commercialization for high-value biomedical applications. These include Terra Verdae Bioworks (Edmonton, AB, Canada), Tepha Inc. (Lexington, MA, USA), and PolyFerm Canada (Kingston, ON, Canada). Their products range from heart valves, scaffolds, biodegradable sutures, and materials for controlled delivery. These product sectors are smaller in volume but have high profit margins. Further discussion of these companies and their activities is given below.

Danimer Scientific (previously MHG), USA, produces PHA under tradename of Nodax™ ( Bainbridge, GA, USA). They entered the PHA space in 2007 by purchasing intellectual property from Procter and Gamble Co (Cincinnati, OH, USA). Danimer produced a copolymer consisting of 3HB units and (R)-3-hydroxyalkanoate comonomer units with mcl-PHA repeat units [257]. Nodax™ PHA is 100% renewable and is produced from readily available feedstock. The company claims that custom formulation of Nodax™ PHA can provide application-specific tailored plastic resins [258].

PHB industrial (Brazil), produced and marketed biodegradable plastic, PHB and P(3HB-*co*-3HV) under the trade name of Biocycle^®^. The production of these polymers was inaugurated in September 2000 by PHB Industrial (Sao Paulo, Brazil) [259]. The industry operated on a pilot scale until 2015 as well as exporting products to Japan. However, the industrial plant is now defunct.

The mission of Bio-on, founded in 2007, was to be integrated with the agri-food industries that they believed would provide the technologies needed to produce PHAs. Unfortunately, Bio-on announced bankruptcy in 20 December 2019 [260]. Bio-on then founded LUX-ON whose focus was to develop PHA manufacturing using carbon dioxide as the feedstock. Their technology also harvests renewable solar energy to power bioproduction process. Bio-on bioplastic currently uses feedstocks that include, sugar cane molasses, sugar cane, food wastes, WCO, glycerol and carbohydrates [261].

TianAn Biopolymers (Ningbo, China), Ecomann Biotechnology (Shenzhen, China), and Tianjin GreenBio Materials (Tianjin, China), and were said to possess a cumulative capacity of 15,000 tonnes/year to produce the PHA materials discussed below [262,263].

TianAn Biopolymers (Ningbo, China) is the world’s largest producer (10,000 tons/year) of P(3HB-*co*-3HV), marketed as ENMAT™. They produce the copolymer by fermentation of *C. necator* on D-glucose and propionic acid. They claim that P(3HB-*co*-3HV) serves as the primary component in materials for injection molding, thermoforming, blown films and extrusions. They also supply PHB (ENMAT Y3000 and Y3000P) along with P(3HB-*co*-3HV)/PLA blends (ENMAT F9000P) [262].

Tianjin GreenBio Material Co. is the first company in China to produce 10,000 tonnes of PHA per year. They developed the pellets (SoGreen 2013) for blown film processing that are fully biodegradable. They also developed PHA foam pellets that can be made into fully-biodegradable foams for the food service industry as well as appliance packaging. The PHA they produce consists of P(3HB-*co*-4HB) copolymer [263].

In 2007, Biomer Biotechnology Co. (Germany) manufactured at both the pilot and research scales 10 tons/year in under the name Biomer™ Biopolyester [264]. To the best of our knowledge, the current status of this company is unclear.

Kaneka Corporation (Japan) manufactures P(3HB-*co*-3HHx), under the tradename Kaneka PHBH^®^ and AONILEX^®^, the reported production capacity in 2007 was 100 tons/year. It is produced by a microorganism fermentation process, in which plant oils and its fatty acids are used as the primary raw material. As of 2019, Kaneka Corporation was producing 5000 tons per year [265].

Yield10 was launched by Metabolix, Inc. in 2015 and is traded on the Nasdaq. Yield10 Bioscience produces PHB and its copolymers under the trade name of Mirel™ [266]. Commercialization of Mirel™ resulted when Archer Daniels Midland Company and Metabolix entered into the joint venture called Telles.

Tepha’s TephaFLEX^®^ PHA consists of 4HB units (i.e., P4HB). After purification, P4HB is processed into medical products that include films, sutures and textile products [267]. The hydrolysis of P4HB leads to 4HB that is a normal constituent of the mammalian body. Tepha’s TephELAST is more elastic than TephaFLEX and is also being applied to develop medical devices. These polymers are produced at both pilot and research scales.

Mitsubishi Gas Chemical (Japan) produces P(3HB) under the trade name Biogreen^®^ at both research and pilot scales [268].

Polyferm Canada produces mcl-PHA under the tradename of VersaMer™ from naturally selected organisms and feedstocks such as vegetable oils and sugars [269]. They are currently developing applications for medical devices, sealants, adhesives, plastic additives and more.

Metabolix, a Massachusetts-based company, completed the sale (US$10 million) of its intellectual property concerning polyhydroxyalkanoate (PHA) biopolymers, to an affiliate firm CJ CheilJedang Corp (Seoul, South Korea). The sale includes production and application patents as well as microbes used in Metabolix’s production processes [270].

Newlight Technologies, using a microorganism isolated from the Pacific Ocean, developed a proprietary biocatalyst for PHA production from carbon dioxide and methane. Newlight takes methane from a dairy farm in California that it mixes with air to form a polymer they trademarked as AirCarbon [271,272]. Information on the production capacity and composition of PHA produced by Newlight technologies is currently not available. However, as a validation of the potential of Newlight technologies, Newlight signed a deal in 2015 with Vinmar International, a petrochemical distributor [273]. Vinmar agreed to purchase over the next 20 years 450 million tons of AirCarbon PHA. Furthermore, once Newlight constructs its production facility (23 million tons/year), Vinmar has committed to purchase 100% of the AirCarbon PHA produced at this site.

## 9. Downstream Processing of PHA

Downstream processing constitutes a critical step in PHA manufacturing with regards to product purity, cost and environmental impact. There are two basic approaches to PHA recovery after fermentation: (i) dissolving the biomass with acids, alkali, surfactants, and enzymes leaving the granules for isolation or (ii) direct solvent extraction of PHAs from the cells that produce it [195].

Methods for the release of PHA granules from cells that largely circumvent solvent utilization, such as enzymatic digestion of non-PHA biomass, are of great interest. With enzyme digestion and other methods that seek to remove biomass from granules, one must consider the extent that outer granule components are removed. Indeed, it is well known that specific proteins are localized on the PHB granule surface. Such proteins include enzymes that function for PHA synthases, PHA depolymerization, polymer surface-displayed proteins (e.g., phasins) and other proteins that regulate granule subcellular localization [274]. In selecting or developing PHA isolation methods, one must balance economic, safety, environmental impact, energy input, recovery yield, PHA purity and ease of scale-up [275]. Other factors include the PHA producing strain, PHA composition and effects on PHA molecular weight [276]. For example, the PHA composition will influence its solubility in solvents (e.g., mcl-PHAs are highly soluble in numerous solvents including acetone where PHB is not). Another important factor in PHA granule isolation is that the methods used must cause minimal change in the polymers’ molecular weight. This is due to that, if the molecular weight of the final melt-processed product falls below the critical chain entanglement molecular weight, the product’s mechanical properties will be compromised. Different downstream processing strategies for PHA isolation/purification are depicted in Figure 8.

### 9.1. Solvent Extraction

This is a common method used due to its ease of operation and simplicity. Solvents rupture the cell increasing cell membrane permeability. Subsequently, solvent access to PHA granules results in solubilization of the polymer. To isolate the polymer, it is precipitated using a non-solvent. Acetone is a preferred solvent for mcl-PHA extraction while chilled methanol or ethanol are preferred non-solvents. Solvent extraction is advantaged in that it results in high-purity PHA with little or no change in PHA molecular weight [277].

#### 9.1.1. Halogenated Solvents

Examples of halogenated solvents used for PHA extraction include 1,2-dichloroethane, methylene chloride and chloroform [278]. Studies show that PHA recovery yields as well as PHA purity are high using these solvents [279]. The most used solvent for extraction is chloroform that, unfortunately, is toxic and increases process cost. PHAs from chloroform extraction have low contents of endotoxin, which is a high priority for PHAs used in medical applications. Several countries have banned the use of chlorinated solvents in consumer products.

#### 9.1.2. Non-Halogenated Solvents

Considering the problems encountered using chloroform or other halogenated solvents, several commercial producers have published patents on chlorinated solvent alternatives [280,281,282,283,284,285,286]. While non-halogenated solvents for polymer extraction may be less harmful to personnel carrying out PHA purification, their sustainability requires careful life-cycle analysis. Also, the inherent process requirements including solvent recycling requires technoeconomic analysis.

Extraction of scl-PHA that, due to their structures (i.e., copolyesters) have low crystallinity, may result in acetone, ethyl acetate, methyl-isobutyl ketone, and cyclohexanone. Extraction of mcl-PHAs can be accomplished at room temperature using diethyl ether, tetrahydrofuran and acetone [276].

As mentioned above, an important consideration during PHA isolation, especially where the final application is as a medical product, is the extent that endotoxins (e.g., lipopolysaccharides) are effectively removed. One approach that has shown promise is extraction methods that are temperature-controlled as demonstrated by Ferrur et al., for the recovery of P(3HO-*co*-3HHx) from *P. putida* GPo1 [287]. Solvents such as 2-propanol and *n*-hexane are particularly useful, giving PHA purities >97% (*w*/*w*) and low endotoxin levels (i.e., 10–15 endotoxin units, EUs) per g PHO. Furthermore, redissolution of PHO at 45 °C in 2-propanol and subsequent precipitation of PHO at 10 °C gave a high purity product (nearly 100%) with correspondingly low endotoxicity (2 EU/g PHO) [287].

Koller et al., explored the potential of applying elevated temperatures and pressures to enhance the ability of acetone to extract scl-PHAs. The results of this approach were compared with solvent extraction at ambient temperature and pressure with chloroform. Extraction of scl-PHA using acetone performs similarly to chloroform in terms of extraction yield (96.8% by both methods) and scl-PHA purity (98.4 vs. 97.7%) [276]. However, extraction using acetone is much more rapid than using chloroform (20 min vs.12 h).

Non-halogenated solvents including acetic acid, methanol, hexane, dimethyl formamide, dimethyl sulfoxide and ethylene carbonate were assessed for PHB extraction from *C. necator* cells [278]. Maximum PHB recovery yield and product purities (98.6 and 98%, respectively) was achieved by using ethylene carbonate at 150 °C for 60 min [278]. Some solvents function to extract lipids from cell debris [288].

#### 9.1.3. Green Solvents

Green processes for PHA extraction will require low energy input, mild conditions, avoid toxic chemicals, and deliver PHA products in high purity and extraction yield [289]. Advantaged solvents include dimethyl carbonate that is fully biodegradable and those that can be produced from biomass and/or bioprocesses from readily renewable resources. Examples of the latter include ethanol and esters of acetate and lactate.

Yabueng et al., explored the use of green solvents for PHB recovery from *C. necator* strain A-04 [290]. Solvents studied included ethyl lactate, 1,3-dioxolane, 2-methyltetrahydrofuran and 1,3-propanediol. The results obtained were compared with that by chloroform extraction. Water was mixed with 1,3-dioxolane to give a water-miscible system. Dried *C. necator* cells containing PHB were extracted at 80 °C for 6 h in 1,3-dioxolane. Subsequently, added water resulted in the phase separation of P3HB and 1,3-dioxolane. This method resulted in isolation of 97.9% ± 1.8% pure PHB with a 92.7% ± 1.4% recovery yield [290]. Analysis of recovered PHB showed that its molecular weight was unaffected.

Characteristic features of ionic liquids are their low vapor pressures and their ability to be tuned to dissolve a wide range of water-insoluble materials [291]. However, one must take care that the produced PHA does not contain residual ionic liquids that may be toxic. Furthermore, recyclability of the ionic liquid is required for economically viable processes.

Dubey et al., reported ionic liquid-based extraction of PHB from *Halomonas hydrothermalis* [292]. The ionic liquid, 1-ethyl-3-methylimidazolium diethylphosphate (EMIM DEP), due to its strong H-bond basicity (*β* = 1.07), easily dissolved *Halomonas hydrothermalis* without water removal (i.e., wet biomass). EMIM DEP disrupts the cell membrane leading to PHB solubilization. Subsequently, PHB was separated by precipitation using methanol as the non-solvent. However, the recovery of PHB was low (60%) as was the percent purity (86%) [292]. Other process concerns are the ability to sufficiently clean and recover EMIM DEP for re-use as it is expensive.

### 9.2. Ultrasound-Assisted Extraction

To reduce PHA recovery cost, it is essential to reach high PHA recovery yields and PHA purities in short time periods while minimizing solvent use [293]. Ultrasound irradiation increases the efficiency of mass transfer during extraction processes. Ishak et al., applied ultrasound irradiation to mcl-PHA containing cells that were suspended in a good and marginal non-solvent mixture [293]. The term marginal non-solvent is meant to describe a substance that, when alone, functions as a non-solvent. However, when that solvent is mixed with a good solvent (i.e., when alone it dissolves the PHA) in a suitable ratio, the PHA remains in solution [294]. By increasing the concentration of the marginal solvent, the PHA will precipitate. These workers interrogated the effects of sonication volumetric energy dissipation, solvent/marginal non-solvent ratio, and time on mcl-PHA extraction efficiency. By choosing heptane as the marginal non-solvent, the application of ultrasound resulted in improved PHA extraction yield. By optimizing process conditions, ultrasound irradiation did not negatively affect mcl-PHA molecular weight and efficiently removes endotoxins [277].

Ultrasound-assisted extraction processes were shown to provide higher extraction yields relative to conventional processes. Furthermore, the application of ultrasound during PHA extraction allows the reduction of solvent quantities, the use of safe solvents and decreased process times [295].

### 9.3. Supercritical Fluid Extraction

Supercritical fluids (SCFs) are substances at a pressure and temperature that is above its critical point. Under these conditions is any substance at a temperature and pressure that is above its critical point, where separate gas and liquid phases are non-existent. Advantageously, diffusion and solvation of SCFs have properties akin to a gas and liquid, respectively. Similar to a gas, SCFs can diffuse through materials and as well as function as liquids by dissolving polymers [279].

The most frequently used SCF is that of carbon dioxide (sCO_2_). Work has demonstrated the ability of sCO_2_ to function in biomolecule purification and separation processes. Further advantages of sCO_2_ is that it is non-toxic, non-flammable and inert chemically [296]. Also, sCO_2_ functions at moderate temperatures (31 °C) and pressures (74 bar). Removal of sCO_2_ subsequent to its use is accomplished by vaporization at reduced pressure. This technique causes cell disruption that allows isolation of PHA in purities that vary between 86 and 99% purity. Raza et al., claimed that, under optimized conditions, supercritical fluid extraction is highly effective providing impurity-free PHAs for medical applications [297]. Supercritical CO_2_ provides a green solvent alternative that deserves further investigation to optimize process efficiency [298]. Primary disadvantages of this method include high capital and maintenance costs.

Hampson and Ashby used sCO_2_ to isolate/purify mcl-PHAs produced by *Pseudomonas resinovorans*. The process used consists of treating the dry cell biomass with sCO_2_ to extract lipid impurities which accounted for 2–11% of the non-PHA biomass [299]. However, these authors then turned to chloroform for isolation/purification of the mcl-PHA. The resulting extraction yield was relatively low (42.4%) [299]. However, the introduction of an initial sCO_2_ step did decrease the quantity of chlorinated solvent used in the second step.

Hejazi et al., also explored the potential use of sCO_2_ for separation of non-PHA biomass from the product. They varied multiple process parameters that include temperature, pressure, time and even use a co-solvent modifier. The optimized process was conducted at 40 °C, 200 atm and 100 min using low quantities of methanol. PHB recovery from *R. eutropha* cells reached 89% [300]. This percent recovery approaches that of other methods described above while it has an improved environmental footprint.

### 9.4. Aqueous Two-Phase Extraction

Purification by an aqueous two-phase extraction (ATPE) involves the formation of two coexisting immiscible phases when one polymer and an inorganic salt are mixed beyond the critical concentration in water. Special features of ATPE is that it consists of higher water content (up to 90% *w*/*w*) affording environmentally friendly conditions for biopolymer separation. Moreover, the ATPE phase-forming components can be non-toxic and compare well to the solvents used in conventional extraction processes. In addition, ATPE reduces the number of purifications step and has high purification capabilities. ATPE has been claimed to provide a readily scalable high-efficiency economic process for PHA purification [13,301,302].

Leong et al., reported optimization of PHB extraction from *R. eutropha*. Using PEG 8000/sodium sulphate at pH 6 and 0.5 M NaCl resulted in a recovery yield of 65% [301]. To enhance the biopolymer recovery, cloud-point extraction (CPE) was used [302]. This allowed reuse of the phase-forming component by using a thermo-responsive polymer (i.e., ethylene oxide-propylene oxide, EOPO). EOPO consists of polyethylene oxide (PEO) and polypropylene oxide (PPO) blocks that are most often synthesized to give PPO-PEO-PPO triblocks or PEO-PEO deblocks. CPE of PHB from *R. eutropha* cells was performed with 20% *w*/*w* EOPO (Mol. wt. 3900 g/mol), NaCl (10 mM) and a phase-separation temperature of 60 °C. The recovery yield was 94.8% with a 1.42 purification factor [302]. The authors demonstrated the EOPO 3900 could be reused at least two times without a significant effect on the recovery yield or the purification factor. Subsequently, Leong et al., reinvestigated the conditions for isolation/purification of PHB from *C. necator* [303]. The process was modified such that the wt-% of EOPO 3900 and ammonium sulfate were 14 and 14, respectively, at pH 6. Under these conditions, extra centrifugation steps were circumvented and the recovery yield and purification factor reached 72.2% and 1.61-fold, respectively [303].

Subsequently, Leong et al., adopted an ATPE process [304]. As above, the removal of non-PHA biomass for to isolate/purify a PHA focused on PHB from *C. necator* H16. Optimized process conditions used were EOPO 3900 (5%), pH 6 and a fermentation temperature of 30 °C. The authors reported a purification factor and recovery yield of 1.36-fold and 97.6%, respectively [304].

### 9.5. Enzymatic and Chemical Digestion Method

The objective of enzyme and chemical digestion processes is to conserve intact PHA granules by dissolution of non-PHA cell mass (NPCM). Approaches have focused on enzymatic and chemical (acid/alkali) processes. The principle of these approaches is to disrupt microbial cell walls and thereby release PHAs from cells [305,306].

Early work explored removal of NPCM by treating cells with strong oxidizing agents such as sodium hydroxide and sodium hypochlorite. Critical to this approach is careful control of oxidizing agent concentration, temperature and reaction time as extreme conditions result in oxidation of both NPCM and PHAs (i.e., reduced molecular weight). Given the lack of selectivity by strong oxidants, research focused on increasingly selective agents such as acids, anionic surfactants and proteolytic enzymes.

Dong et al., studied combinations of surfactant and sodium hypochlorite for recovery of PHA from *Azotobacter chroococcum* G-3 [307]. The biomass was first pretreated by freezing which liberated the PHA granules that were isolated by centrifugation. Another approach combined sodium hypochlorite and surfactant treatments. In one example, PHA-containing cells were incubated for 15 min with sodium dodecyl sulfate 10 g/L (SDS). This treatment solubilized the lipids and proteins. Subsequently, incubation of the NPCM with sodium hypochlorite (30%, 3 min) eliminated peptidoglycans and other impurities. The combined treatment with surfactant and sodium hypochlorite resulted in a PHA recovery and a final purity of 86.6% and 98%, respectively [307].

Proteolytic enzymes catalyze hydrolytic reactions that target proteins. In 2006, Kapritchkoff et al., used enzymes to lyse *R. eutropha* DSM545 and thereby purify PHB [308]. Preliminary studies revealed that the most promising enzymes for this process are lysozyme, bromelain, and trypsin. The best result used bromelain (14.1 U/mL) at pH 9 and 50 °C which gave PHB in 88.8% purity [308]. Subsequently, pancreatin was assessed for cell lysis and removal of NPCM. This resulted in isolation of 90% pure PHB with a 3-fold reduction in cost relative to bromelain and no loss in PHB molecular weight [308]. The selectivity and mild conditions by which enzymes function provide important advantages relative to other methods that makes it likely that enzymes will continue to be evaluated and process improvements will be made [309]. It also may be that combinations of enzymes as well as enzyme-surfactant systems may further boost the efficiency of PHA purification processes.

Alcalase and lysozyme are both effective in digesting cellular biomass. Yasotha et al., attempted to optimize mcl-PHAs recovery/purification by digesting denatured proteins with alcalase (a protease), SDS to enhance solubilization of NPCM, EDTA (Ethylenediaminetetraacetic acid) for divalent metal complexation and lysozyme to catalyze the decomposition of peptidoglycans in cell walls [310]. Analysis of experiments revealed that alcalase contributed most (71% relative to other process factors) to NPCM breakdown and mcl-PHA isolation/purification. Crossflow ultrafiltration effectively removed NPCM and facilitated mcl-PHA granule recovery (90% efficiency) and in high purity (92.6%) [310].

Kachrimanidou et al., reported the isolation/purification of P(3HB-*co*-3HV) by an enzyme process. They used solid-state fermentation of the *Aspergillus oryzae* to develop a mixture of crude enzymes to catalyze *C. necator* lysis to liberate the PHA copolyesters. The enzymatic process was conducted at 48 °C without pH control. The results of this work are impressive as poly(3HB-*co*-3HV) recovery yield and purity reached 98% and 96.7%, respectively [311].

To isolate/purify from *C. necator* cells amorphous PHB granules, Martino et al., used alcalase (0.3 AU/g), EDTA (0.01 g/g), and SDS (0.3 g/g). P3HB granules were recovered with purities >90% without crystallization [312]. In other words, amorphous PHB granules formed during cultivation remained amorphous under the imposed recovery conditions.

Israni et al., harnessed the high lytic activity of *Streptomyces albus* Tia1 for isolation/purification of PHA from PHA-producing cells [313]. In one approach, *S. albus* and PHA-producing cells were co-inoculated. In the second approach, the lytic enzymes from *S. albus* were introduced at the end of PHA cultivations for PHA recovery. Conditions resulting in the maximum activity for cell lysis of PHA were the addition of a concentrated *S. albus* culture filtrate (33.3 mL) from a 100 mL cultivation to *B. megaterium* (220 mg) with incubation at 40 °C and pH 6. By co-inoculation of *S. albus* and PHA producing *B. megaterium*, the PHA yield reached 0.55 g/g which was similar to treatment with sodium hypochlorite for PHA recovery [313]. In contrast, comparison of the enzyme based with co-inoculation showed that the former resulted in a 1.74-fold increase in PHA yield [313]. The authors attributed this difference to use by *S. albus* of released PHA as a carbon source.

Although enzymatic digestion methods are selective, energy efficient and result in good recovery, it is generally agreed that improvement in process economics is required [288].

### 9.6. New Biological Recovery Methods

Less traditional methods for PHA biological recovery include the use of insects and animals. These methods eliminate the use of solvents and hazardous chemicals resulting in ecofriendly processes. The bases for these approaches are that insects and animals will utilize lyophilized bacterial biomass but not PHA. Murugan et al., reported that the larval form of the mealworm beetle, *Tenebrio molitor*, can selectively consume NPCM of *C. necator* without breaking down PHA granules that it excretes in its feces [314]. The resulting feces containing PHA were purified using detergent, water and heat giving PHA granules that are almost 100% pure. Furthermore, the molecular weight of the larva-processed isolated PHA was identical to the same product isolated by chloroform extraction [314]. Nevertheless, there are questions about the scalability of such processes.

Kunasundari et al., reported P3HB isolation using laboratory rats [315]. *C. necator* H16 cells where P3HB was 39% of its CDW was fed to rats. The test rats excreted pellets of PHB with 82–97% purity in the form of feces [315]. The polymer so obtained was further purified using detergent, water and heat. The result was highly purified P3HB granules. Molecular weights of rat-processed P3HB was nearly identical to that isolated by chloroform extraction [315].

## 10. Conclusions: Challenges and Future Perspectives

PHAs have a long history of study both by science/engineering academicians, industrial, and national laboratory researchers who have been working towards successful commercialization of PHA-based materials. These bioplastics offer an appealing platform of polymer compositions that can be integrated into biorefinery processes. It is this vision that has resulted in a history of over 80 years of fundamental and applied studies as well as multiple investments for scale-up and commercialization of PHA products.

The present report describes PHA biosynthesis strategies to obtain desired copolymer compositions where process efficiency was emphasized. The design of microbial fermentation processes depends on the production organism, feedstock selection as well as the cultivation conditions. Implementation of metabolic engineering and evolutionary approaches will lead to continual improvements in PHA production efficiency, copolymer structural diversity and cost-performance metrics. Process development must carefully consider the PHA production organism, consistent availability of feedstocks and metrics such as energy input, carbon conversion efficiency, substrate cost, cell growth rate and PHA production rate. While PHA titer and productivity are important measures of success, many published works do not provide carbon utilization rates which is a critical determinant of a successful outcome. Furthermore, techno-economic analysis and life-cycle analysis are rarely included in published works.

The development of organisms that are metabolically engineered to better produce PHAs from waste materials such as molasses, whey, spent cooking oils, and various hydrolyzed lignocellulosic raw materials will likely play a critical role in the development of robust and cost-effective processes. While much progress has been made, metabolic engineering can avoid the use of feedstocks such as γ-butyrolactone, levulinic acid and fatty acids to produce copolymers with 4HB, 3HV and 3HHx units. Metabolic engineering can introduce metabolic routes to desired monomers from unrelated carbon sources. Halophilic PHA producers have great potential as they allow PHA production to be conducted under non-sterile conditions in media resembling seawater. Furthermore, placing these organisms in DI results in cell lysis easing downstream processing. More attention should be paid to increasing the metabolic potential of these strains to produce the desired copolymer products at high titers and productivity.

There is no question that a longstanding Achilles heel to PHA commercialization has been the need for cost-efficient downstream processing methods. Effective methods will likely not be generic to an organism or product but will be customized and optimized for selected systems. Residual impurities in PHAs can result in melt-processed products of reduced molecular weight and coloration. While enzymatic processes have shown promise, their efficiency must be improved to increase PHA purity and reduce enzyme costs. Routes to recycle enzymes used in removing non-PHA biomass are important as a route to reduce costs. The use of impure enzyme cocktails generated by microbes is interesting but challenging since the cocktail composition must be optimized and result in consistent outcomes.

What is beyond the scope of this review is PHA physical–mechanical properties. While its academically interesting to pursue PHAs with new compositions of matter, it is crucial that application scientists focus on the many tools of material science which includes developing optimal blend compositions and composites that improve mechanical and rheological properties while retaining chemical and biological recycling.

Although there are likely many cynics who, based on the commercial history of PHAs believe it foolish to consider further investments in PHA technologies, in the view of the authors of this review biotechnological developments in concert with work by fermentation engineers, materials scientists and breakthroughs in lignocellulose processing will lead to a future where PHAs will provide sustainable commodity materials for a range of applications.

## Figures and Tables

**Figure 1 molecules-26-03463-f001:**
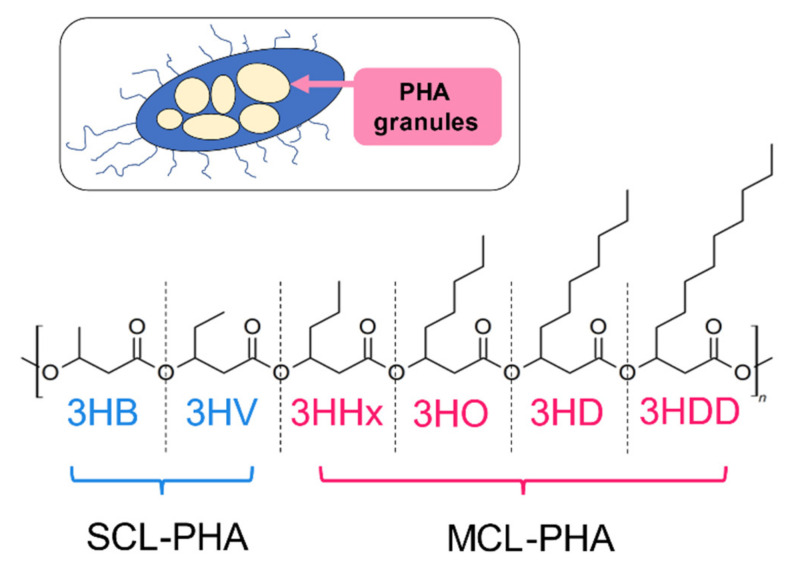
Polyhydroxyalkanoates with short chain-length repeat units (Scl-PHA) monomers include 3-hydroxybutyrate (3HB) and 3-hydroxyvalerate (3HV). Mcl-PHA monomers include 3-hydroxyhexanoate (3HHx), 3-hydroxyoctanoate (3HO), 3-hydroxydecanoate (3HD), and 3-hydroxydodecanoate (3-HDD). PHA granules accumulate within the cytoplasm of the bacteria cell.

**Figure 2 molecules-26-03463-f002:**
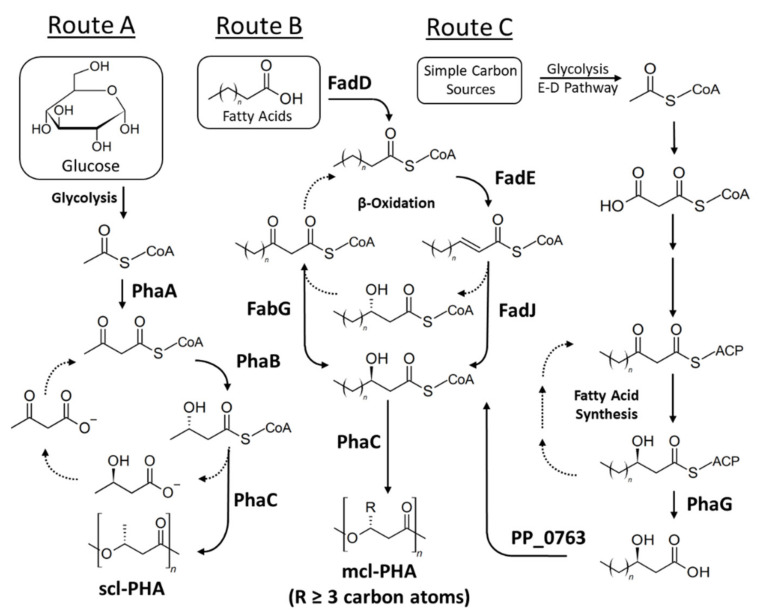
Metabolic routes for PHA biosynthesis.

**Figure 3 molecules-26-03463-f003:**
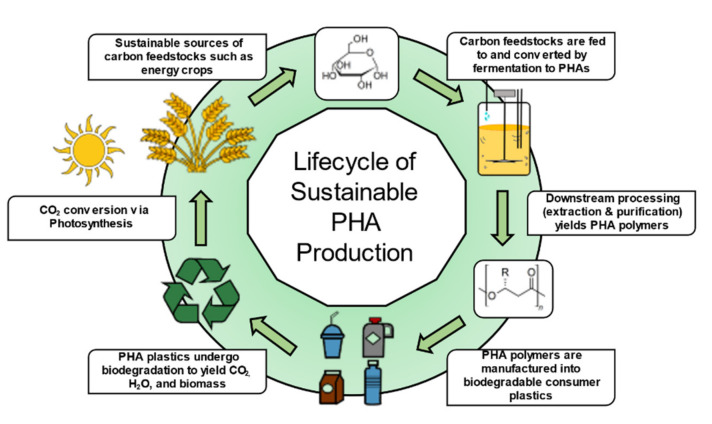
Using renewable, low-cost carbon sources such as energy crops or waste streams in combination with optimized fermentation strategies supports high yielding PHA production processes. The PHA polymers are processed and manufactured into consumer plastics that will biodegrade when disposed. The cycle begins again when biodegradation products of PHA plastics are consumed during photosynthesis or are recovered in waste streams from composting facilities.

**Figure 4 molecules-26-03463-f004:**
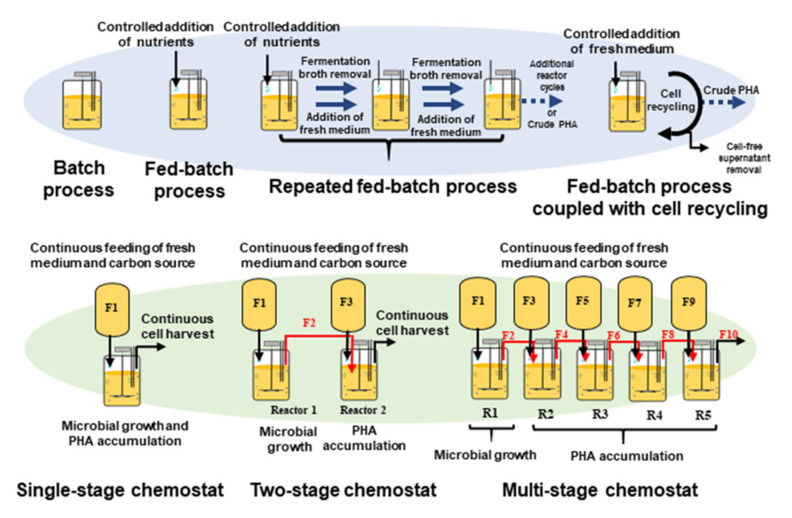
Process regimes for PHA fermentative synthesis.

**Figure 5 molecules-26-03463-f005:**
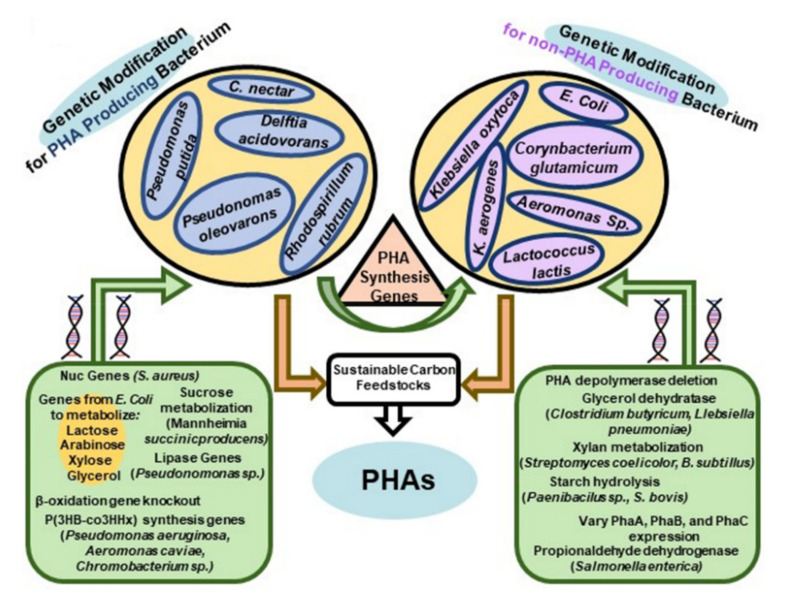
Genetic engineering strategies to improve bacterial strains for PHA production. PHA producing strains can be improved to convert various carbon substrates into PHAs via gene insertion or deletion. PHA synthase genes from PHA producing strains are often inserted in non-PHA producing strains. Using different strains of bacteria facilitates PHA production under conditions that are not suitable for naturally producing PHA strains.

**Figure 6 molecules-26-03463-f006:**
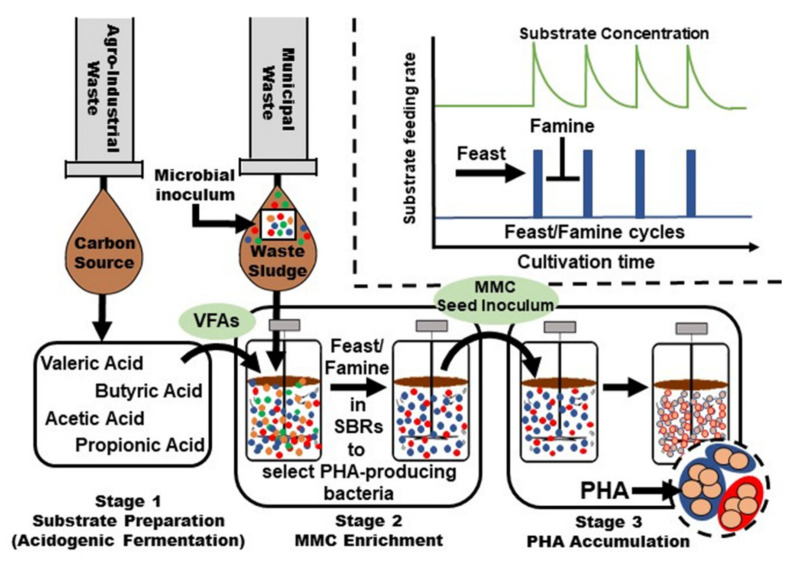
The use of mixed microbial consortia (MMC) for PHA production.

**Figure 7 molecules-26-03463-f007:**
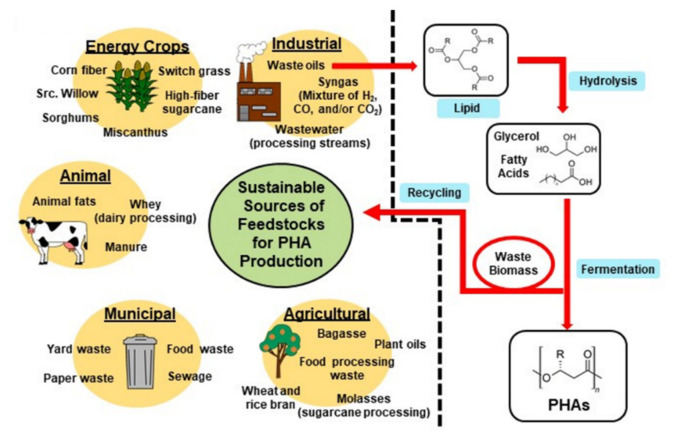
Sustainable sources and conversion of feedstocks for PHA production. Provided within the dashed area is a simplified representation of the conversion of waste oils (lipids) to PHAs. The waste lipid is hydrolyzed to yield substrates for fermentation: fatty acids and glycerol. The substrates are purified prior to use in fermentations. Recovered waste biomass can be recycled as a carbon source for future fermentations.

**Figure 8 molecules-26-03463-f008:**
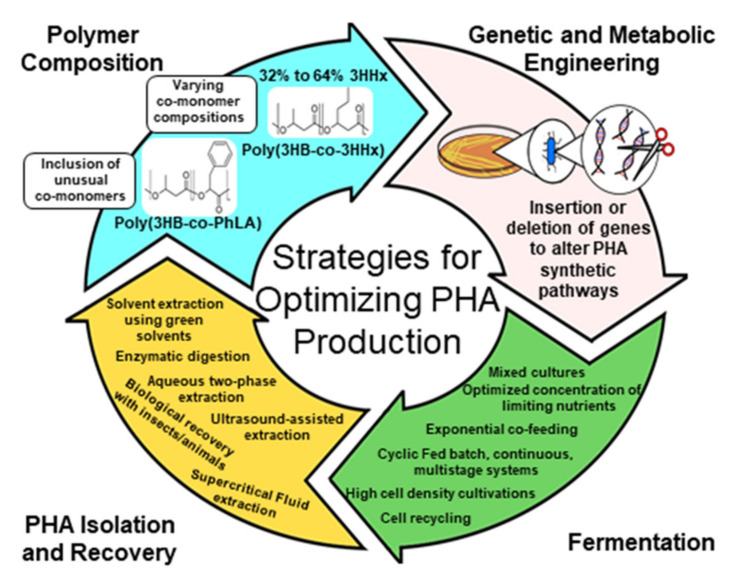
Key areas of PHA production to optimize to obtain high-yields of PHA. To compete with petroleum-based plastics whilst remaining as environmentally friendly as possible, PHA production requires constant innovation and optimization in four major areas. The cyclic arrows illustrate that any modification to one of the four categories will affect the following stage of production. The strategies presented herein have been proposed due to the published success regarding improved PHA production yields or PHA properties.

**Table 1 molecules-26-03463-t001:** PHA production by high cell-density cultivation (HCDC) fed-batch operations.

Microorganisms	Stages of Fermentation	PHA Type	Carbon Source	Biomass (g/L)	PHA Content(%)	Overall Productivity(g/L/h)	Carbon Conversion Efficiency (Y_p/s_) g/g	Reference
*P. putida* KT2440ATCC 47054	One stage	PHO/PHD	Oleic acid	141	51	1.91	NA	[15]
		PHN	Glucose/ nonanoic acid	102	32	0.97	0.56	[26]
		PHN	Nonanoic acid	109	63	2.3	NA	[27]
		PHD	Decanoic acid/ glucose	75	74	1.16	0.86	[55]
		PHN	Glucose/ nonanoic acid	71	56	1.44	0.66	[56]
		PHN	Glucose/ nonanoic acid	56	67	1.44	0.60	[58]
	Two stage	mcl-PHA	Oleic acid/linoleic acid/palmitic acid/stearic acid	125	54.4	1.01	0.70	[62]
*R. eutropha* H16	Two stage	P(3HB-*co*-3HV)	Acetic acid/ propionic acid/butyric acid	112	83	2	NA	[28]
*Alcaligenes latus* DSM 1122	One stage	PHB	Sucrose	143	50	3.97	NA	[40]
*Cupriavidus necator* DSM 545	Two stage	PHB	Glucose	164	76.2	2.03	NA	[41]
	One stage	PHB	Glucose	148	76	3.1	0.33	[43]
		PHB	Glucose/ Fructose	61.6	68.8	1.0	NA	[45]
		P(3HB-*co*-3HV)	Glucose/ propionic acid	80	73	1.24	NA	[46]
		PHB	Glycerol	68.56	64.55	0.76	0.34	[47]
*Burkholderia sacchari* DSM 17165	One stage	P(3HB-*co*-4HB)	Saccharose/*γ*-butyrolactone	77	72.6	1.87	0.275	[42]
		PHB	Glycerol	43.79	10.22	0.08	0.41	[47]
*Zobellella denitrificans* MW1	NA	PHB	Glycerol	82.2	67	1.09	NA	[48]
*C. necator* A-04	One stage	P(3HB -*co*-4HB)	Fructose/1,4 butanediol	112	65	0.76	NA	[49]
*Halomonas venusta* KT832796	One stage	PHB	Glucose	38	88	0.25	0.22	[51]
*Burkholderia sacchari* IPT 189	One stage	P(3HB-*co*-3HV)	Sucrose/ propionic acid	NA	60	1.04	0.25	[52]
*Aeromonas hydrophila* 4AK4	One stage	P(3HB-*co*-3HHx)	Glucose/ lauric acid	50	50	0.54	NA	[53]
*P. putida* LS46	One stage	PHO	Octanoic acid	29	61	0.66	0.62	[54]
*P. putida* CA-3	Two stage	PHD	Butyric acid/decanoic acid	71.3	65	1.63	0.55	[57]
*P. putida* IPT046	One stage	mcl-PHA	Glucose/fructose	50	63	0.80	0.19	[59]
*P. oleovorans* ATCC 29347	One stage	PHO	Octanoic acid	63	62	1	NA	[60]
*P. putida* BM01	Two stage	PHO	Octanoic acid/glucose	55	65	0.90	0.40	[61]
*Cupriavidus necator* MTCC 1472	One stage	PHB	Sucrose	37.56	61.82	0.58	0.20	[63]
*Azohydromonas* *lata* DSM 1123	One stage	PHB	Sucrose	-	94	4.2	0.15	[64]

Y_p/s_: PHB yield to substrate (g PHB produced per g substrate consumed); %-PHA: Final intracellular PHA content (g/g) of DCW; NA: Data not available, PHB: Polyhydroxybutyrate, P(3HB-*co*-3HV): Poly (3-hydroxybutyrate-co-3-hydroxyvalerate), PHO: Polyhydroxyoctanoate, PHD: Polyhydroxydecanoate, PHN: Polyhydroxynonanoate, P(3HB-*co*-4HB): Poly (3-hydroxybutyrate-co-4-hydroxybutyrate), P(3HB-*co*-3HHx): Poly(3-hydroxybutyrate-co-3-hydroxyhexanoate), P(3HB -*co*-4HB): Poly (3-hydroxybutyrate-co-4-hydroxybutyrate)

**Table 2 molecules-26-03463-t002:** PHA production by HCDC continuous mode operations without cell recycle.

Carbon Source	Microorganisms	Stages of Fermentation	Biomass (g/L)	%PHA	Overall Productivity (g/L/h)	Carbon Conversion Efficiency (Y_p/s_) g/g	Reference
*n*-Octane	*P. oleovorans* ATCC 29347	Two stage	18	63	1.06	NA	[65]
Glucose	*C. necator* DSM 545	Five stage	81	77	1.85	NA	[66]
	*C. necator* DSM 545	Multi stage	80	NA	2.14	0.47	[67]
	*R. eutropha* WSH3	Two stage	50	73	1.23	0.36	[68]
	*R. eutropha* WSH3 (lab collection)	Two stage	32.6	75	NA	0.043	[69]
	*Halomonas* TD01	One stage	20	65	0.26	0.242	[75]
	*Chelatococcus* sp. Strain MW10	semi-continuous/cyclic fed-batch	115	12	NA	0.09	[78]
Fructose	*Wautersia eutropha* NRRL B-14690	Two stage	49	51	0.42	NA	[70]
Butyric acid and valeric acid	*R. eutropha* DSM 428	One stage	NA	40	NA	NA	[72]
Glucose/ sodium propionate	*R. eutropha* (ATCC 17699, CCRC 13039)	One stage	7.96	30	0.045	NA	[73]
Oleic acid	*P. putida* KT2442	Two stage	30	23	0.69	0.15	[74]
Sucrose	*Azohydromonas lata*	Two stage	20.52	83.43	NA	NA	[79]

Y_p/s_: PHB yield to substrate (g PHB produced per g substrate consumed); %-PHA: Final intracellular PHA content (g/g) of DCW; NA: Data not available.

**Table 3 molecules-26-03463-t003:** Accumulation of PHA produced by different strains under nitrogen/phosphorus/oxygen limitation conditions.

Microorganisms	PHA Type	Limiting Nutrients	Carbon Source	Biomass (g/L)	%-PHA	Overall Productivity (g/L/h)	Carbon Conversion Efficiency (Y_p/s_) g/g	Reference
*P. putida* KT 2440 ATCC 47054	mcl-PHA	Phosphorus (4 g/L)	Oleic acid	141	51.4	1.91	NA	[15]
		Phosphorus (22 g/L)		173	18.7	1.13		
*Alcaligenes eutrophus* NCIMB 11599	PHB	Phosphorus (5.5 g/L)	Glucose	281	82	3.14	0.38	[29]
*R. eutropha* NCIMB 11599	PHB	Phosphate	Glucose	208	67	3.1	NA	[39]
Activated sludge	PHB/PHV	Phosphorus	Acetate	NA	37	NA	NA	[85]
		Nitrogen		NA	59	NA		
*C. necator* DSM 545	P(3HB-*co*-3HV)	Phosphorus	Propionic acid/butyric acid	65.9	88	0.65	0.51	[87]
*Burkholderia* *sacchari* IPT 189	PHB	Nitrogen/oxygen	Sucrose	150	42	1.7	0.22	[88]
*C. necator* DSM545	PHB	Phosphate	Butyric acid	46.7	82	0.57	0.62	[89]
*Methylobacterium* *organophilum* NCIB 11278	PHB	Potassium	Methanol	250	52	1.86	0.19	[90]
*Ralstonia eutropha* ATCC 17699	P(3HB-*co*-4HB)	Nitrogen	Fructose + γ-butyrolactone	48.5	50.2	0.55	NA	[91]

Y_p/s_: PHB yield to substrate (g PHB produced per g substrate consumed); %-PHA: Final intracellular PHA content (g/g) of DCW; NA: Data not available, PHB: Polyhydroxybutyrate, PHV: Polyhydroxyvalerate, P(3HB-*co*-3HV): Poly (3-hydroxybutyrate-co-3-hydroxyvalerate), P(3HB-*co*-4HB): Poly (3-hydroxy-butyrate-co-4-hydroxybutyrate).

**Table 4 molecules-26-03463-t004:** PHA production by recombinant starins under HCDC conditions.

Microorganisms	PHA Type	Carbon Source	Biomass (g/L)	%-PHA	Overall Productivity (g/L/h)	Carbon Conversion Efficiency (Y_p/s_) g/g	Reference
Recombinant *E. coli*	P4HB	Glycerol/acetate/4-hydroxy-butyrate	43.2	33	0.207	NA	[50]
	PHB	Lactose	194	87	4.6	0.45	[92]
	PHB	Glycerol	42.9	63	0.45	NA	[109]
	PHB	Glucose/ thiamine	156	72	2.4	NA	[112]
	PHB	Lactose	87	80	1.4	0.11	[113]
	P(3HB-co-3HV)	Glucose/ oleic acid/ propionic acid	42.2	70	1.37	0.5	[128]
Recombinant *R. eutropha*	P(3HB-co-3HHx)	Palm oil	139	74	1.06	0.52	[93]
	P(3HB-co-3HHx)	Waste animal fat	45	60	0.4	0.40	[117]
	P(3HB-co-3HHx)	Soybean oil	133	72.5	1	0.74	[121]
*E. coli* strain K1060	PHB	Corn steep liquor/milk whey	70.1	73	2.13	NA	[110]
*Escherichia coli* XL1-Blue	PHB	Glucose	178	72	NA	NA	[111]
Recombinant *C. necator* H16	P(3HB-co-3HHx)	Palm kernel oil/butyrate	171	78	-	NA	[116]
Recombinant *Aeromonas hydrophila* 4AK4	P(3HB-co-3HHx)	Dodecanoate	54	52.7	0.791	NA	[124]
	P(3HB-co-3HHx)	Dodecanoate: sodium gluconate	38.4	52	0.475	NA	
Recombinant *C. necator*	P(3HB-co-3HHx)	Sludge palm oil	88.3	57	1.1	0.7	[126]
Recombinant *Halomonas Bluephagenesis* TD01	P(3HB-co-4HB)	Glucose, γ-butyro- lactone	83	60.62	1.04	0.27	[129]
Recombinant *P. putida* KT2440	PHD	Glucose	62	67	0.83	NA	[130]
Recombinant *E. coli* XL1-Blue	P (PhLA-co-3HB)	Glucose	25.27	55	0.145	NA	[131]

Y_p/s_: PHB yield to substrate (g PHB produced per g substrate consumed); %-PHA: Final intracellular PHA content (g/g) of DCW; NA: Data not available, P4HB: Poly(4-hydroxybutyrate), PHB: Polyhydroxybutyrate, P(3HB-co-3HV):Poly(3-hydroxybutyrate-co-3-hydroxyvalerate), P (PhLA-co-3HB): Poly(Phenyllactate-co-3-hydroxybutyrate), P(3HB-co-3HHx): Poly(3-hydroxybutyrate-co-3-hydroxyhexanoate), P(3HB-co-4HB): Poly(3-hydroxybutyrate-co-4-hydroxybutyrate).

**Table 5 molecules-26-03463-t005:** Production of different types of PHA polymers by recombinant strains.

Microorganisms	Polymer Type	PHAs Composition	Carbon Source	Reference
Recombinant *Escherichia coli*	Polylactic acid random copolymer	Poly(lactate-co-3-hydroxybutyrate)	Xylan	[17]
Recombinant *Pseudomonas putida*	Block copolymer	Poly(3-hydroxybutyrate-b-poly4-hydroxybutyrate)	sodium butyrate/γ-butyrolactone	[100]
Recombinant *Escherichia coli*	Block copolymer	poly(3-hydroxybutyrate)-b-poly(3-hydroxypropionate)	Glycerol	[107]
Recombinant *P. entomophila* LAC23	Homopolymer	Poly(3-hydroxyheptanoate) Poly(3-hydroxyoctanoate) Poly(3-hydroxynonanoate) Poly(3-hydroxydecanoate) Poly(3-hydroxyundecanoate) Poly(3-hydroxytetradecanoate) Poly(3-hydroxytridecanoate)	Sodium heptanoate/ sodium octanoate/ sodium nonanoate/decanoic acid/undecanoic acid/dodecanoic acid/tridecanoic acid/tetradecanoic acid	[133]
Recombinant *P. entomophila* LAC23	Random copolymer	P(3hydroxyoctanoate-co-3hydroxydodecanoate)	Sodium octanoate/dodecanoic acid	[133]
		P(3hydroxyoctanoate-co-3hydroxytetradecanoate)	Sodium octanate/ tetradecanoic acid	
*P. entomophila* LAC32	Diblock copolymer	P(3-hydroxyoctanoate)-b-P(3-hydroxydodecanaote)	Sodium octanoate/dodecanoic acid	[133]
Recombinant *Pseudomonas entomophila* LAC23	Homopolymer	Poly(3-hydroxy-9-decenoate)	9-decenol	[134]
Recombinant *Pseudomonas entomophila* LAC23	Block copolymer	P(3-hydroxydodecanoate-b-3-hydroxy-9-decenoate)	dodecanoic acid/9-decenol	[134]
Recombinant *Pseudomonas putida*	Homopolymer	Poly(3-hydroxyhexanoate) Poly(3-hydroxyheptanoat) Poly(3-hydroxyoctanoate-co-2 mol% 3-hydroxyhexanoate) Poly(3-hydroxyvalerate) Poly(3-hydroxybutyrate) Poly(4-hydroxybutyrate)	Hexanoate/ Heptanoate/ Octanoate/Valerate/γ-butyrolactone	[135]
Recombinant *Pseudomonas putida*	Homopolymer	Poly(3-hydroxydecanoate) Poly(3-hydroxydecanoate-co-84 mol%3-hydroxydodecanoate)	Decanoic acid/Dodecanoic acid	[136]
Recombinant *E. coli*	Random copolymer	P(3-hydroxybutyrate-co-3-hydroxypropionate)	Glucose	[137]
Recombinant *Pseudomonas putida*	Random copolymer	Poly(3-hydroxybutyrate-co-3-hydroxyhexanoate-co-3-hydroxyoctanoate)	Sodium heptanoate/ Oleic acid	[138]
*P. entomophila* LAC32	Random copolymer	P(3-hydroxybutyrate-co-3-hydroxydecanaote)	Glucose/ decanoic acid/dodecanoic acid	[139]
		P(3-hydroxybutyrate-co-3-hydroxydodecanaote)		
*Pseudomonas putida* KT2442	Random copolymer	Poly (3-hydroxybutyrate-co-3-hydroxyhexanaote)	Sodium butyrate/ Sodium hexanoate	[140]
*Pseudomonas putida* KT2442	Random copolymer	Poly (3-hydroxybutyrate-co-mcl 3HA)	scl-fatty acid/mcl-fatty acid	[141]
*P. mendocina* NK-01	Random copolymer	P(3-hydroxyoctanoate-co-3-hydroxydecanoate-co-3-hydroxydodecanoate), P(3-hydroxyhexanaote-co-3-hydroxyoctanaote-co-3-hydroxydecanaote-co-3-hydroxydodecanoic acid)	sodium octanoate/Sodium decanoate/dodecanoic acid	[142]
Recombinant *Pseudomonas entomophila* LAC23	Random copolymer	P (3-hydroxydodecanoate-co-3-hydroxy-9-decenoate)	dodecanoic acid/9-decenol	[144]
Recombinant *Pseudomonas putida*	Block copolymer	Poly(hydroxybutyrate-b-polyhydroxyvalerate-hexanoate-heptanaote	Butyrate/hexanaote	[144]
	Random copolymer	Poly(3-hydroxybutyrate-co-valerate-hexanoate-heptanaote)	Butyrate/hexanaote	
*Ralstonia eutropha* NCIMB 11599	Triblock copolymer	Poly(3-hydroxybutyrate-co-3-hydroxyvalerate-b-poly(3-hydroxybutyrate)-b- Poly(3-hydroxybutyrate-co-3-hydroxyvalerate)	Glucose/Pentanoic acid	[145]
*R. eutropha* NCIMB 11599	Block copolymer	Poly(3-hydroxybutyrate-co-3-hydroxyvalerate)-b-poly(3-hydroxybutyrate)	Glucose/Pentanoic acid	[146]
Recombinant *Pseudomonas putida* KT2442	Diblock copolymer	Poly(3-hydroxybutyrate -b-poly-3-hydroxyhexanoate)	Sodium butyrate/sodium hexanoate	[147]
*Burkholderia sacchari* DSM 17165	Block copolymer	Poly(3-hydroxybutyrate-b-3-hydroxyvalerate)	Xylose/levulinic acid	[148]
Mixed culture containing *Azohydromonas lata* DSM 1122 and *Burkholderia**sacchari* DSM 17165	Random copolymer and block copolymer	Poly(3-hydroxybutyric-co-3-hydroxyvalerate-co-4-hydroxyvalerate) P(3-hydroxybutyrate-b-3-hydroxyvalerate)	Glucose/levulinic acid	[150]
*Pseudomonas putida* Gpo1	Functional polymer (cationic PHA)	Poly[(*β*-hydroxy-octanoate)-co-(*β*-hydroxy-11-(bis(2-hydroxyethyl)-amino)-10-hydroxyundecanoate)]	Sodium octanoate	[151]
Recombinant *Escherichia coli* and *Pseudomonas putida*	Functional polymer	3-hydroxydecanoic acid	Fructose	[152]
Recombinant *Escherichia coli*	Polylactic acid random copolymer	P(lactic acid-co-3-hydroxybutyrate-co-3-hydroxypropionate)	Glucose/glycerol	[161]
Recombinant *Escherichia coli*	Polylactic acid random copolymer	Poly(glycolate-co-lactate-co-3-hydroxybutyrate)	Glucose	[162]
Recombinant *Escherichia coli*	Polylactic acid random copolymer	Poly(glycolate-co-lactate-co-3-hydroxybutyrate-co-4- hydroxybutyrate)	Glucose	[163]
Recombinant *Escherichia coli*	Polylactic acid random copolymer	Poly (lactate-co-3- hydroxybutyrate)	Xylose/acetate	[164]
Recombinant *Escherichia coli*	Polylactic acid random copolymer	Poly(lactate-co-3-hydroxybutyrate)	Xylose-based hydrolysate	[165]
Recombinant *Escherichia coli*	Polylactic acid random copolymer	Poly(D-lactate-co-glycolate-co-4-hydroxybutyrate)	Glucose/xylose	[166]
Recombinant *Escherichia coli*	Polylactic acid random copolymer	Poly(lactate-co-3-hydroxybutyrate)	Glucose	[167]

Y_p/s_: PHB yield to substrate (g PHB produced per g substrate consumed); %-PHA: Final intracellular PHA content (g/g) of DCW; NA: Data not available.

**Table 6 molecules-26-03463-t006:** Production of scl- and mcl-PHA using renewable resources.

Microorganisms	PHA Type	Carbon Source	Biomass (g/L)	%-PHA	Overall Productivity (g/L/h)	Carbon Conversion Efficiency (Y_p/s_) g/g	Reference
*Cupriavidus necator* DSM 545	P(3HB-co-4HB)	Waste glycerol/γ-butyrolactone	30.19	36.1	0.17	0.06	[23]
	P(3HB-4HB-3HV)	Waste glycerol/γ-butyrolactone/propionic acid	45.25	36.9	0.25	0.08	
*Bacillus megaterium* BA-019	PHB	Sugarcane molasses	72.6	42.1	1.27	NA	[38]
*C. necator* DSM 545	PHB	Waste glycerol	104.7	62.7	1.36	NA	[41]
*Zobellella denitrificans* MW1	PHB	Crude glycerol	81	66.9	1.09	0.25	[48]
*Burkholderia sacchari* DSM 17165	PHB	Wheat straw hydrolysate	146	72	1.6	0.22	[211]
*Burkholderia sacchari* DSM 17165	P(3HB-co-4HB)	Wheat straw hydrolysate/-butyrolactone	88	27	0.5	NA	[212]
*Wautersia eutropha* NCIMB 11599	PHB	Wheat hydrolysate	175.05	93	0.89	0.47	[217]
*Paraburkholderia sacchari* IPT 101 LMG 19450	PHB	Hardwood hydrolysate	59.5	58	0.72	0.15	[219]
*Cupriavidus necator* DSM 4058	PHB	Crude glycerol/rapeseed meal	28.86	85.75	0.21	0.32	[226]
*Paracoccus* sp. LL1	P(3HB-co-3HV)	Crude glycerol	24.2	39.3	NA	0.136	[227]
*Cupriavidus eutrophus* B-10646	PHB	Glycerol (99.3% purity) (30 L fermenter)	69.3	72.4	NA	NA	[228]
		Glycerol (99.7% purity) (30 L fermenter)	69.4	73.3	NA	NA	
		(Glycerol 82.1% purity) (30 L fermenter)	69.3	78.1	NA	NA	
		Glycerol (99.7% purity) (150 L fermenter)	110	78	1.83	NA	
*Burkholderia cepacia* ATCC 17759	PHB	Crude glycerol	23.6	31	NA	NA	[229]
*C. necator* DSM 7237	PHB	Crude glycerol and sunflower meal hydrolysate	37	72.9	0.28	0.32	[230]
	P(3HB-co-3HV)	Sunflower meal hydrolysate and levulinic acid	35.2	66.4	0.24	0.28	
*P. fluorescens* A2a5	PHB	Cane liquor medium	32	68.75	0.23	NA	[233]
*Bacillus* *megaterium* BA-019	PHB	Sugarcane molasses	90.71	45.85	1.73	NA	[234]
*Cupriavidus necator*	PHB	Sugarcane vinnase and molasses	20.89	56	NA	NA	[237]
*R. eutropha* NCIMB 11599	PHB	Waste potato starch	179	52.51	1.47	0.22	[239]
*R. eutropha* NCIMB 11599	PHB	Wheat bran hydrolysate	24.5	62.5	0.25	0.32	[243]
*Haloferax mediterranei* ATCC 33500	P(3HB-co-3HV)	Extruded rice bran/extruded cornstarch/yeast extract	140	55.6	3.2	NA	[244]
*H. mediterranei* ATCC 33500	P(3HB-co-3HV)	Enzymatic extruded starch	39.4	50.8	0.29	NA	[244]
*P. aeruginosa* STN-10	P(3HB-co-3HV)	Waste frying oil	44.71	53	0.33	NA	[247]
*Cupriavidus necator* H16	P(3HB-co-3HV)	Waste rapeseed oil/propanol	138	76	1.46	0.83	[248]
*Cupriavidus necator* DSM 428	PHB	Used cooking oil (Exponential profile)	21.3	84	0.1875	0.65	[249]
		Used cooking oil (DO stat strategy)	27.2	77	0.525	0.52	
*Pseudomonas* *putida* KT2440ATCC 47054	PHD/PHO	Hydrolyzed waste cooking oil	159.3	36.4	1.93	0.76	[250]
*P. chlororaphis* 555	PHD/PHDD/PHO/PHHx	Waste cooking oil	73	19	0.29	0.11	[251]
*Salinivibrio* sp. M318	PHB	Waste fish oil and glycerol	61.1	51.5	0.46	0.32	[252]
*C. necator* DSM 545	PHB	Soybean oil	83	80	2.5	0.83	[253]
*P. putida* KT2442	PHO/PHD	Corn oil hydrolysate	103	27.1	0.61	NA	[254]

Y_p/s_: PHB yield to substrate (g PHB produced per g substrate consumed); %-PHA: Final intracellular PHA content (g/g) of DCW; NA: Data not available, PHB: Polyhydroxybutyrate, P(3HB-co-4HB): Poly(3-hydroxybutyrate-co-4-hydroxybutyrate), P(3HB-co-3HV): Poly (3-hydroxybutyrate-co-3-hydroxyvalerate), P(3HB-4HB-3HV): Poly(3-hydroxybutyrate-4-hydroxybutyrate-3-hydroxyvalerate), PHO: Polyhydroxyoctanaote, PHD: Polyhydroxydecanoate.

**Table 7 molecules-26-03463-t007:** Pilot and industrial-scale PHA manufacturers currently active worldwide.

Company	Trade Name	PHA Type	Raw Material	Production Capacity (tons per year)	Price per kg	References
Danimer Scientific (Bainbridge, GA, USA)	Nodax™	PHBH	NA	20	NA	[257,258]
PHB Industrial (Sao Paulo, Brazil)	Biocycle^®^	PHB, P(3HB-*co*-3HV)	Sugarcane	600	NA	[259]
Bio-On, (Bologna, Italy)	Minerv-PHA™	PHA	Sugar beet and sugar cane molasses, other waste-derived feedstocks.	10,000	NA	[261]
TianAn GreenBio Materials Co. Biopolymer, (Ningbo, China)	ENMAT™, SoGreen™	PHB, P(3HB-*co*-4HB)	Dextrose	10,000	NA	[262]
GreenBio-DSM (TEDA Tianjin, China)	Ecoflex blend Enmat^®^	P(3HB-*co*-4HB), P(3HB-*co*-3HV) + Ecoflex blend Enmat^®^	NA	10,000	€3.26	[263]
Biomer Biotechnology Co. (Schwalbach, Germany)	Biomer^®^ biopolyesters	PHB	Glucose from corn starch	50	€3.00–5.00	[264]
Kaneka Corporation (Minato-ku, Tokyo, Japan)	PHBH™	P(3HB-*co*-HHx)	Vegetable oil	5000	NA	[265]
Yield10 Bioscience, (formerly Metabolix, Inc.), (Woburn, MA, USA)	Mirel™	PHB copolymers (e.g., P(3HB-*co*-HHx-*co*-HO)	Corn sugar	50,000	NA	[266]
Tepha Inc (Lexington, MA, USA)	TephaFlex^®^ TephElast^®^	P4HB, P3HB-*co*- 4HB	NA	NA	NA	[267]

NA: Data not available.

## Data Availability

Data supporting reported results can be found in publicly archived datasets such as SciFinder and Scopus.

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
