# Peer review of "Emergent Approaches to Efficient and Sustainable Polyhydroxyalkanoate Production"

_molecules, 2021, doi:10.3390/molecules26113463_

Round 1

Reviewer 1 Report

The review contains useful information in the field of biopolymer. I suggest the authors to address following comments before further consideration.

  • Please further suggest current challenges for the production of PHA, and how to overcome such challenges.
  • What are next efforts in academic researches and industries to make the PHA production process more commercially viable than current situration.
  • Please introduce more works published in 2021.

Author Response

  1. Please further suggest current challenges for the production of PHA, and how to overcome such challenges.
  2. What are next efforts in academic researches and industries to make the PHA production process more commercially viable than current situration.

For 1 and 2, we have provided this information in the Review. It is given throughout the manuscript for specific examples of hurdles/opportunities and at the end in the newly named section Conclusions: challenges and future perspectives. 

Examples of text throughout the Review that addresses future challenges/opportunities are below:  

Current challenges: See page 2, lines 55-71 A major bottleneck in the commercialization of PHAs is high cost of production where metrics such as carbon conversion yield (g/g), titer or volumetric yield (g/L) and productivity (g/L/h) are critically important [11]. In addition to production metrics, low cost downstream processing methodologies and PHA manufacturing that meets cost-performance requirements have remained challenging. This has inspired researchers to work towards increasing PHA fermentation and downstream processing efficiency to reduce the overall cost [12-15].

6.1. Lignocellulosic feedstock

A great opportunity exists to exploit the estimated 200 billion tons global production of lignocellulosic feedstocks [202,203]. It’s well known that efficient utilization of lignocellulosic feedstocks comes with numerous challenges since these materials were designed by nature to be difficult to access by microbes to enable their long lifetime as plant structural materials.

 Jung et al. reported a continuous two-stage process by which P. oleovorans converts n-octane to mcl-PHA. Two stage fermentations offer the opportunity to focus on biomass accumulation in the first bioreactor and PHA accumulation in a second bioreactor. In the first (D1) and second (D2) bioreactors, the dilution rate were 0.21 h-1 and 0.16 h-1, respectively. These conditions resulted in a total biomass, %-mcl-PHA content in the CDW and overall productivity of 18 g/L, 63% and 1.06 g/L/h, respectively [64].

Pichia pastoris has proved highly useful for the high-level expression of heterologous proteins [154]. Furthermore, P. pastoris has proved amenable to genetic manipulation. Also, this organism naturally synthesizes mcl-PHAs [155]. Vijayasankaran et al. reported that by the introduction of R. eutropha PHA biosynthesis genes, the recombinant P. pastoris strain showed an enhanced ability to accumulate P3HB [156]. This provides the opportunity to co-express the formation of PHAs and high-value proteins, an approach that can improve PHA economics. 

See: Conclusions: challenges and future perspectives

See Figure 8 and corresponding discussions.

8.1.3. Green solvents

Green processes for PHA extraction will require low energy input, mild conditions, avoid toxic chemicals, and deliver PHA products in high purity and extraction yield [277]. Advantaged solvents include dimethyl carbonate that is fully biodegradable and those that can be produced from biomass and/or bioprocesses from readily renewable resources. Examples of the latter include ethanol and esters of acetate and lactate. 

3. Please introduce more works published in 2021.

This manuscript was completed just prior to 2021.  It covers the literature up through December 31 2020

Reviewer 2 Report

The review paper entitled “Emergent Approaches to Efficient and Sustainable Poly-hydroxyalkanoate Production” is the best structurally arranged paper I have had the opportunity to review. What I would point out as the strongest side of this review paper is that it emphasizes production of poly-hydroxyalkanoate. Modes of operations are discussed in detail as well as effects of nutrient limitations on yield of PHA and represent the basis for bioprocess engineers who consider this production from both professional and scientific aspects. The review of PHA production using genetically modified organisms was also excellently done. Different types of industrial / agro-industrial waste are analyzed in detail as potential raw materials in this production, in accordance with the principles of circular economy. With the presented overview of global PHA producer companies at pilot and industrial scale and downstream processing of PHA, this thematic is completely rounded off. I recommend this paper for publication in the presented form and, when it is publicly available, I will be happy to recommend it to doctoral students in biotechnology as an example of a quality review paper and as a great basis for considering the topic of PHA production.

Author Response

The authors appreciate your enthusiasm for the Review. We are also excited to be able to provide this resource to the community 

Reviewer 3 Report

Manuscripts is a review dealing with different aspects of PHA production. Document contains a massive amount of information, that is in general terms well organized and presented. Manuscript deals with a wide variety of factors related with PHA production, and a lot of data is nicely presented in a series of tables. Therefore, it is inferred that manuscript, if published, may turn into a useful document for potential readers. However, in my opinion, few things need to be addressed by authors before publication is advised.

Introduction section is not very well organized. It does not provide a general view that efficiently introduce the reader to the subject. Some paragraphs have little connection with the others (for example lines 84-88). Revision of that section in required. I wonder if Figure 2 is really needed, from my perspective, it does not add-up anything useful for the introduction.

Some of the figure legends are excessively long. For example, the ones of Figure 4 and 6, which actually contains information that should be in the text of the manuscript. Indeed, many legends not only describe the figures, but also provide comments or discussion that should not bet there.

Most of the document is organized following this structure, per each subject: General description of the subject, followed by a series of paragraphs describing one research each. In principle there is nothing wrong with that. However, since information is also presented in tables, sometimes information is provided twice. Authors are suggested to provide as much information as possible in their tales, leaving the text for comments and discussion. This will help to reduce the length of the documents, which is I think too long.

Conclusion section is little long, and a big part of it may be used to structure a General Discussion Section. So, I would suggest the authors to analyze the possibility to include a Discussion section, where they can try to “wrap-up” the information of the paper, identifying trends or common ideas, and suggestions. Then, Conclusions may be a more direct and concise section.

Check lines 74-75, text “as to: 1)” should be removed.

Author Response

As suggested by the Reviewer, we have re-organized the introduction.

Figure 2. We believe this Figure provides fundamentals that will provide the reader with the fundamental background so they can more readily understand the discussions below on relationships between carbon sources and PHA composition as well as basic pathway genetic manipulations.  

As suggested by the Reviewer, Text from long Figure legends was moved into the main manuscript.

We believe that, for the convenience of readers, having data in the text is important since it avoids the constant need to go between the Tables and the text.   Also, listing of data in Tables has a different purpose as it gives readers an opportunity to compare results under different production conditions.  Therefore, we see both as essential.  

Given there is a massive amount of information in this Review, we used the Conclusions as a way to discuss current challenges and future perspectives.